# Individual and healthcare supply-related HIV transmission factors in HIV-positive patients enrolled in the antiretroviral treatment access program in the Centre and Littoral regions in Cameroon (ANRS-12288 EVOLCam survey)

**Pierre-julien Coulaud**[1]*, **Abdourahmane Sow**[1], **Luis Sagaon-Teyssier**[1], **Khadim Ndiaye**[1], **Gwenaëlle Maradan**[1,2], **Christian Laurent**[3], **Bruno Spire**[1], **Laurent Vidal**[1], **Christopher Kuaban**[4], **Sylvie Boyer**[1], for the EVOLCam Group¶

**1** Aix Marseille Univ, INSERM, IRD, SESSTIM, Sciences Economiques & Sociales de la Santé & Traitement de l'Information Médicale, ISSPAM, Marseille, France, **2** ORS PACA, Observatoire Régional de la Santé, Provence-Alpes-Côte d'Azur, Marseille, France, **3** IRD, INSERM, Univ Montpellier, TransVIHMI, Montpellier, France, **4** Faculté de Médecine et de Sciences Biomédicales, Université de Yaoundé 1, Yaoundé, Cameroun

¶ Membership of the EVOLCam Group is listed in the Acknowledgments.
* pierre-julien.coulaud@inserm.fr

## Abstract

### Background

Despite great progress in antiretroviral treatment (ART) access in recent decades, HIV incidence remains high in sub-Saharan Africa. We investigated the role of individual and healthcare supply-related factors in HIV transmission risk in HIV-positive adults enrolled in 19 HIV services in the Centre and Littoral regions of Cameroon.

### Methods

Factors associated with HIV transmission risk (defined as both unstable aviremia and inconsistent condom use with HIV-negative or unknown status partners) were identified using a multi-level logistic regression model. Besides socio-demographic and behavioral individual variables, the following four HIV-service profiles, identified using cluster analysis, were used in regression analyses as healthcare supply-related variables: 1) district services with large numbers of patients, almost all practicing task-shifting and not experiencing antiretroviral drugs (ARV) stock-outs (n = 4); 2) experienced and well-equipped national reference services, most practicing task-shifting and not experiencing ARV stock-outs (n = 5); 3) small district services with limited resources and activities, almost all experiencing ARV stock-outs (n = 6); 4) small district services with a wide range of activities and half not experiencing ARV stock-outs (n = 4).

### Results

Of the 1372 patients (women 67%, median age [Interquartile]: 39 [33–44] years) reporting sexual activity in the previous 12 months, 39% [min-max across HIV services: 25%-63%]

**Data Availability Statement:** Fully anonymized data are available on request from the study investigators (Laurent Vidal at laurent.vidal@ird.fr and Christopher Kuaban at ckuaban@yahoo.fr).

**Funding:** This study was funded by the French National Agency for Research on HIV/AIDS and viral hepatitis (ANRS, grant 12288, PIs: LV and KB, https://www.anrs.fr/fr). PJC was the recipient of a doctoral fellowship from ANRS (B7-ANRS 12324, https://www.anrs.fr/fr) and Sidaction (17-2-FJC-11561, https://www.sidaction.org/). The funders had no role in study design, data collection and analysis, decision to publish, or preparation of the manuscript.

**Competing interests:** The authors have declared that no competing interests exist.

**Abbreviations:** AIC, Akaike's Information Criterion; aOR, adjusted Odds Ratio; ART, Antiretroviral Treatment; ARV, Antiretroviral drugs; CI, Confidence Interval; HIV, Human Immunodeficiency Virus; HSP, HIV-Service Profiles; ICC, intra-class correlation coefficient; IQR, Interquartile Range; LMIC, Low- and Middle-Income Countries; MCA, Multiple Correspondence Analysis; MD, Missing data; PLHIV, People Living with HIV; STIs, Sexually Transmitted Infections; UNAIDS, Joint United Nations Programme on HIV/AIDS; VLS, Viral Load Suppression; WHO, World Health Organization.

were at risk of transmitting HIV. The final model showed that being a woman (adjusted Odd Ratio [95% Confidence Interval], p-value: 2.13 [1.60–2.82], p<0.001), not having an economic activity (1.34 [1.05–1.72], p = 0.019), having at least two sexual partners (2.45 [1.83–3.29], p<0.001), reporting disease symptoms at HIV diagnosis (1.38 [1.08–1.75], p = 0.011), delayed ART initiation (1.32 [1.02–1.71], p = 0.034) and not being ART treated (2.28 [1.48–3.49], p<0.001) were all associated with HIV transmission risk. Conversely, longer time since HIV diagnosis was associated with a lower risk of transmitting HIV (0.96 [0.92–0.99] per one-year increase, p = 0.024). Patients followed in the third profile had a higher risk of transmitting HIV (1.71 [1.05–2.79], p = 0.031) than those in the first profile.

## Conclusions

Healthcare supply constraints, including limited resources and ARV supply chain deficiency may impact HIV transmission risk. To reduce HIV incidence, HIV services need adequate resources to relieve healthcare supply-related barriers and provide suitable support activities throughout the continuum of care.

## Introduction

Despite great progress in implementing innovative HIV programs over the last two decades, HIV incidence remains high in many countries worldwide [1]. In 2020, 1.5 million people were newly infected [2], which is four times the UNAIDS target of a maximum of 370,000 annual new HIV infections by 2025 [3].

There is wide consensus on the effectiveness of combined HIV prevention initiatives, which comprise various behavioral, biomedical and structural interventions, to prevent HIV transmission (2). One key biomedical intervention is early antiretroviral treatment (ART) initiation which has been shown to dramatically reduce HIV-related mortality and morbidity as well as HIV transmission risk [4–7]. The beneficial effect of early ART on viral load suppression (VLS) led to the establishment of the U = U ("undetectable equals untransmittable") movement, which is widely recognized for its importance in controlling the HIV epidemic [8]. This treatment as prevention approach evolved into the test-and-tread strategy, that is to say ART initiation immediately after HIV diagnosis, irrespective of CD4 count [9].

In order to enhance efforts to reduce HIV transmission, in 2014 UNAIDS launched the "90-90-90" care continuum targets which aimed to diagnose HIV infection in 90% of all people living with HIV (PLHIV), initiate ART in 90% of diagnosed PLHIV, and achieve VLS in 90% of PLHIV treated with ART, all by 2020 [10]. In 2021, a new Global AIDS Strategy was adopted with the aim of reaching "95-95-95" targets by 2025 [11].

However, several barriers continue to hamper the achievement of these targets in low- and middle-income countries (LMIC), including weaknesses in healthcare supply in health systems, primarily a lack of equipment and of human resources [12]. Achieving the third UNAIDS 95 target is particularly challenging in LMIC, especially in West and Central Africa where only 59% of treated PLHIV had achieved VLS in 2020 [13]. In addition, despite the growing need for ART in LMIC, financial support for HIV has substantially decreased in this region (-11% over 2010–2020) [13]. This is particularly worrying for Western and Central African countries, where international support accounted for more than three quarters of total available HIV resources in 2020 [13].

In order to assess the capacity of national programs to reach the third UNAIDS 95 target, it is essential to monitor two key related indicators: VLS and HIV transmission risk. To date, most studies measuring HIV transmission risk in various populations and contexts have used behavioral characteristics as outcome variables (mainly knowledge of one's own HIV status and inconsistent condom use) [14–16]. Only a small number of recent studies have used a more comprehensive approach, that is to say, combining both behavioral characteristics and biomedical (e.g., viral load) factors [6, 17–19]. Furthermore, the literature on the factors associated with HIV transmission risk in Africa focuses primarily on individual factors related to adherence to ART [20–22] and VLS [23–25]. The few studies to date exploring the role of healthcare supply-related factors [24, 26, 27] only assessed a limited number of HIV service characteristics and did not consider the more complex nature of the organization of these services (e.g., dimensions such as the working process, available resources and management of ARV). In a previous study conducted in Cameroon, we showed that HIV service profiles, built using a cluster analysis of a wide range of healthcare supply-related characteristics, had different performances in terms of time to ART initiation [28]. In the present study, we used a similar approach to provide a better understanding of the role of supply-related factors, beside individual factors, on achieving the third UNAIDS target in Cameroon. This study also provided the opportunity to highlight challenges related to the implementation of the 2018–2022 Cameroonian National Strategic Plan for HIV/AIDS and STIs which aims to reduce the number of new HIV infections by 60% and achieve VLS in 92% of PLHIV on ART in Cameroon by 2022 [29].

Accordingly, we aimed to: i) measure the prevalence of HIV transmission risk in HIV-positive patients enrolled in the Cameroonian ART access program in the country's Centre and Littoral regions, and ii) investigate the roles of both individual and healthcare supply-related factors in HIV transmission risk.

## Materials and methods

### Study setting: The national Cameroonian ART program

Cameroon is an LMIC in Central Africa affected by a generalized HIV epidemic with a mean estimated prevalence rate in 2018 of 2.7% in adults (aged 15–49 years), and large disparities according to gender, region and urban area [30]. The highest prevalence rates are observed in women (3.4%), in urban areas (3.9%), and in the South (5.8%), East (5.6%), Adamaoua (4.1%), North-West (4.0%) and Centre (3.5%) [30] regions.

Created in 2001, the national ART access program initially led to the implementation of Accredited Treatment Centers, first in the national reference hospitals in Yaoundé and Douala, and then in the regional hospitals of the country's eight other regions. From 2005 onwards, ART delivery services were extended to the district hospital setting, with the setting up of HIV Management Units. The Cameroon national authorities decided to provide ART for free in 2007 and to remove all user fees for HIV care in 2019 [31]. Thanks to this continuously developing HIV strategy, the total number of PLHIV on ART increased from 17,940 in 2005 to 145,038 in 2014 and 367,871 in 2020 [32, 33].

In its latest National Strategic Plan for HIV/AIDS and STIs for the period 2018–2022 [29], Cameroonian health authorities promote a set of priority strategies and interventions including i) strengthening HIV prevention, ii) scaling-up HIV testing and ART access through the decentralization of HIV care, task shifting, and the involvement of community-based organizations, iii) ensuring the permanent availability of laboratory equipment, antiretroviral drugs and medicines of opportunistic infections.

## Study design and data collection

We used data from the cross-sectional survey EVOLCam (ANRS 12288) which was conducted in 19 HIV services in Cameroon's Centre (n = 11) and Littoral regions (n = 8) between April and December 2014 to study evolutions in the national ART program through a comparison with the 2006–2007 ANRS-12116 EVAL survey [20, 34]. The EVOLCam (ANRS 12288) study protocol is described in detail elsewhere [35, 36].

Briefly, eligible PLWH (≥21 years old and HIV diagnosed >3 months) attending an outpatient consultation in one of the 19 participating HIV services were randomly selected and informed about the study. Patients willing to participate provided written informed consent before data collection. First, a standardized medical questionnaire was completed during the consultation by healthcare providers. The following clinical data were obtained from patient examinations and retrospective medical files: dates of HIV diagnosis and ART initiation, WHO clinical stage of HIV infection at ART initiation and at the time of the study, CD4 count at ART initiation, drug regimen at the time of the study, body mass index and any history of tuberculosis and hepatitis B co-infection and related diagnosis date. Second, patients answered a face-to-face questionnaire administered in a private room by trained independent interviewers which collected data on demographic, socioeconomic, behavioral, psychosocial and domestic information. More specifically, a series of questions were asked on adherence to ART, perceived health and HIV-related stigma as well as alcohol consumption and sexual behaviors during the 12 months prior to the study (number of sexual partners, experience of transactional sex, frequency of sexual relationships, HIV status and condom use with the two most recent partners). The questionnaire is available as supplementary material in Fiorentino et al., 2021 [19]. Third, a blood sample was taken to measure HIV viral load (only for patients ART treated >6 months) and CD4 cell count. All blood samples were analyzed by a reference HIV laboratory in Yaoundé.

Finally, detailed data on the characteristics of the participating 19 HIV services were collected through interviews with hospital staff, *in situ* observations, and cross-validation with data recorded in HIV service activity reports. Specifically, the information obtained included: i) hospital's general characteristics (location, opening date, legal status, number of beds), ii) human resources working in the HIV service (number and qualifications), iii) activity (number of ART-treated patients and available services including educational, nutritional and financial support, HIV community-based organization involvement), iv) HIV service organization (separate ARV storage, stock management and task-shifting for clinical consultations of ART-treated patients and/or ARV prescription renewals), v) technical resources (functional medical imaging equipment, CD4 count machine, ARV stock-outs for at least one of the three most prescribed ART regimens [28].

This study was conducted in compliance with international and national regulations on ethics and research on people. It received administrative authorisation from the Ministry of Public Health in Cameroon and was approved by the Cameroonian National Ethics Committee (approval reference: 2013/08/349/L/CNERSH/SP). All individual data collected in the research were anonymized using a patient identification number; only this number was reported in the data collection tools and the databases used for analyses.

## Study population

The study population for the present analysis comprised participants in EVOLCam who reported having at least one sexual partner in the 12 months prior to the survey and who had no missing data for the variables used to build the study outcome.

## Outcome

The study outcome was a binary variable describing the risk of transmitting HIV (yes *versus* no). No standard method exists to define the risk of HIV transmission. Accordingly, in order to define the outcome, we used the literature to develop a comprehensive approach which included both biomedical (i.e., unstable aviremia) and behavioral factors (i.e., inconsistent condom use with negative or unknown HIV status partner(s)) [6, 17–19]. More specifically, we defined the risk of transmitting HIV as a combination of both unstable aviremia and reporting inconsistent condom use either with the most recent (i.e., in the previous 12 months) sexual partner (if only one partner declared), or with at least one of the two most recent sexual partners (if more than one partner declared) of negative or unknown HIV status.

Unstable aviremia was defined as not currently being treated *or* on treatment for less than six months *or* on treatment for more than six months but with a detectable viral and/or poor adherence to ART [19]. The latter was defined as taking <80% of the prescribed drug doses *or* reporting treatment interruptions for at least two consecutive days in the four weeks prior to the survey [37]. Participants on treatment for more than six months with an undetectable viral load who were highly adherent to ART (defined as taking >80% of the prescribed drug doses in the four weeks prior to the survey) were considered to have stable aviremia.

Inconsistent condom use was defined as answering i) "Never", "sometimes" or "almost always" to the question "In the 12 months prior to this survey, did you use condoms with this partner?" and/or ii) "No" to the question "During your most recent sexual intercourse with this partner, did you use a condom?".

## Explanatory variables

**Individual-related variables.** The following individual variables were considered in the present analysis:

- Socio-demographic and economic characteristics: age, gender, residential setting (urban *versus* rural), marital status (single, married, common law union), having a main partner, living with one's main partner, number of children (0, 1–4, ≥5), currently desiring or trying to have a child, educational level (lower than secondary school *versus* secondary school and above), having an economic activity, and household monthly income.

- Sexual behaviors and psychosocial characteristics: number of partners (in the 12 months prior to the survey and in lifetime), transactional sex (paid for or received), perception of HIV-related stigma (score 0–8 computed using the HIV Stigma Scale [38]), mental quality of life (measured using the SF12 scale [39]) and frequent binge drinking (defined as drinking ≥3 large bottles of beer (i.e., ≥260 cL in total) and/or 6 other alcoholic drinks on one occasion at least once a month).

- Clinical data: time since HIV diagnosis, disease symptoms at HIV diagnosis, time from HIV diagnosis to ART initiation (<2 *versus* ≥2 months), and CD4 cell count at ART initiation (<100, ≥100 cells/mm$^3$, not treated).

**HIV service profile variable.** The above healthcare supply-related characteristics (see subsection 'Study design and data collection') were used to build an HIV service profile variable using a multiple correspondence analysis (MCA) combined with a hierarchical cluster analysis [28]. More specifically, the MCA allowed us to combine healthcare supply-related correlated variables to create continuous factors, which were then used in a classification procedure to identify service profiles (clusters), hereafter called 'HIV-service profiles' (HSP) (see S1–S3 Figs for more details) [40, 41]. The method used is fully described elsewhere [28].

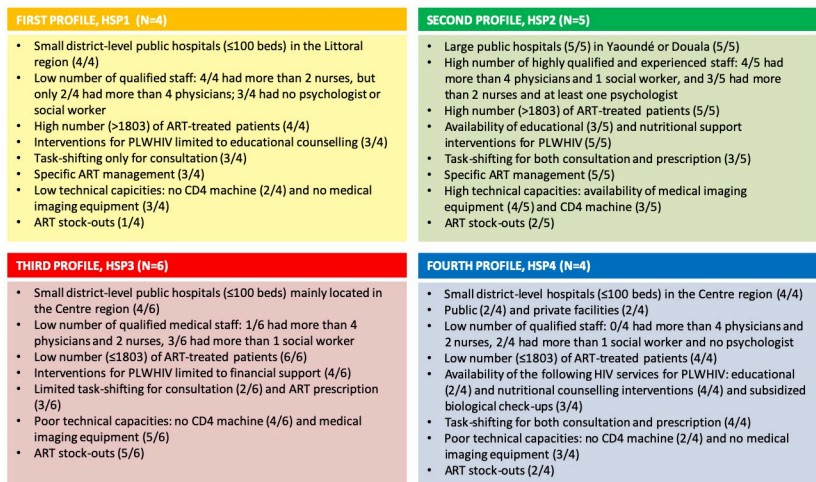

**Fig 1. Main features of each HIV-service profile (EVOLCam survey, ANRS 12288).** Abbreviations: ART = Antiretroviral treatment; ARV = Antiretroviral drugs; PLHIV = People Living with HIV.

This cluster analysis led to the identification of the four following distinct HSP.

- HSP1 (n = 4) included small district HIV services with limited equipment and staff that provided ART to a large number of patients. Most services (3/4) practiced task-shifting and educational counselling, reported using managing ARV stock separately, and did not experience ARV stock-outs.

- HSP2 (n = 5) included experienced and well-equipped national reference services with a high number of ART-treated patients. Most services (3/5) practiced task-shifting. All five used separate stock management, and most (3/5) did not experience ARV stock-outs.

- In HSP3 (n = 6), most HIV services were small district hospitals with limited resources and activities, and a small number of patients. Task-shifting was limited and almost all (5/6) experienced ARV stock-outs.

- HSP4 (n = 4) included small district services, following a small number of patients with limited human and technical resources. However, they provided a wide range of HIV support activities and practiced task-shifting. Half did not experience ARV stock-outs.

Characteristics of the 19 participating HIV services and of the 4 HSP identified are described in detail in S1 Table, while Fig 1 summarizes the main features of each HSP.

## Statistical analysis

We described individual characteristics of the study population, both overall and according to the outcome, using numbers (percentages) for categorical variables and median [interquartile range, IQR] for continuous variables. We also described the variability of the outcome overall across all 19 HIV services, and according to each HIV service and each HSP.

We investigated the factors associated with HIV transmission risk using a multilevel logistic regression model. This allowed us to accurately assess the effects of individual characteristics (level 1) and of structural characteristics, that is to say the healthcare supply-related characteristics described in the four HSP categories above and summarized by the HSP variable (level 2) [42]. The multivariate model was built following the modelling

strategy recommended for multilevel models [43]. Initially, we estimated the empty model (without any explanatory variables) to provide an estimation of the inter-class variance, which was small but significantly different from 0 ($\sigma^2(u_0)$ = 0.09; p = 0.032), confirming the relevance of using a multilevel model. We also computed the estimated intra-class correlation coefficient (ICC), which represents the proportion of the inter-class variance compared to the total variance (i.e., inter- and intra-class variance). It was estimated at 0.027 indicating that 2.7% of the outcome's variance was due to differences between HIV services. Individual-related factors in the level-1 model were then selected using a stepwise backward procedure. Only significant individual variables with a p-value <0.05 were retained, except age which was kept in order to control for key demographic characteristics. Finally, the HSP variable was introduced as a level-2 variable to obtain the final multilevel model. Model fit was assessed using Akaike's Information Criterion (AIC). Analyses were performed using SAS (version 9.4), RStudio (version 1.1.453) and Stata/SE version 14.2 (College Station, Texas, United States).

## Results

### Selection of the study population

Of the 2130 HIV-positive patients enrolled in the EVOLCam survey, 748 (35%) were excluded from the present analysis because they did not report any sexual partner in the 12 months prior to the survey interview. A further 10 (1%) were also excluded because of missing data on viral load and/or ART adherence. The study population therefore comprised 1372 PLHIV reporting at least one sexual partner during the 12 months prior to the survey, and with complete data for the study's outcome (see Fig 2).

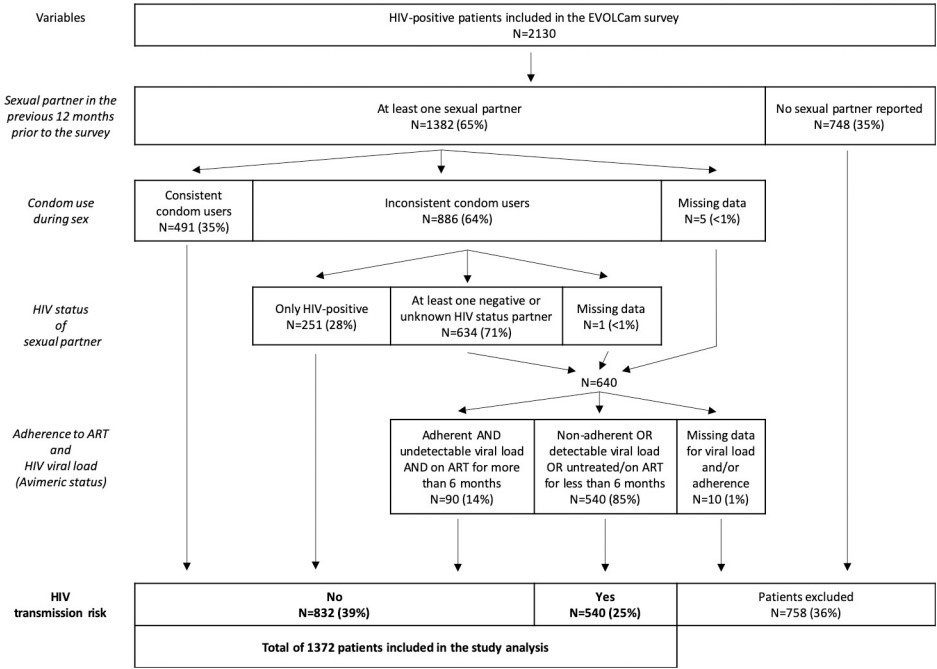

**Fig 2. Flow chart of the study population (selected participants from the EVOLCam survey, ANRS-12288).**

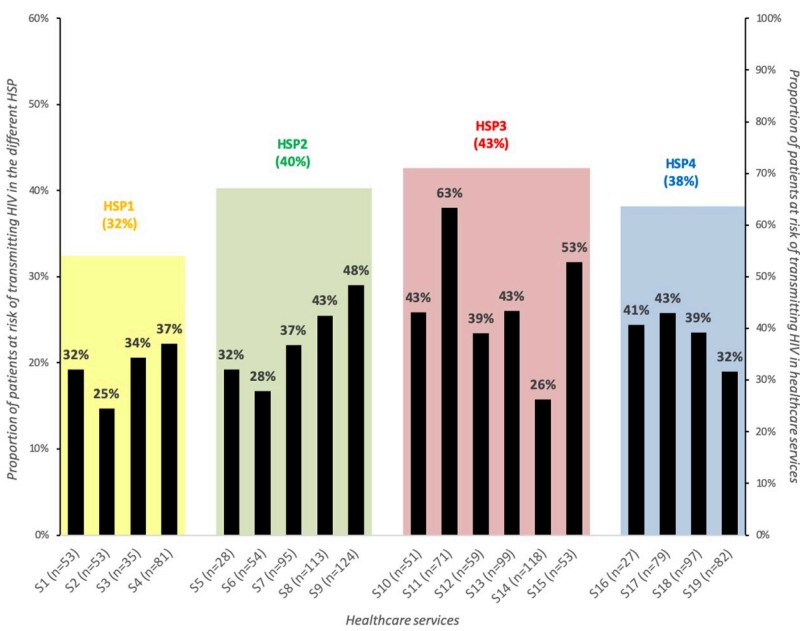

**Fig 3. HIV transmission risk across HIV services and HIV-service profiles (EVOLCam survey, ANRS 12288).** Abbreviations: HSP = HIV service profiles. The names of the participating HIV services are anonymized and numbered from 1 (S1) to 19 (S19).

## Study population characteristics

Patients were predominantly women (67%) and median age [Interquartile (IQR)] was 39 [33–44] years (S2 Table). Most were living in urban areas (85%), had a main partner (86%) and at least one child (89%). A minority (10%) had an educational level equivalent to or higher than secondary school, and approximately two-thirds reported an economic activity. Median [IQR] monthly household income was 15 USD [IQR: 7–29] per adult-equivalent. Further details on sexual behaviors and psychosocial and clinical characteristics are described in S1 Table.

## Descriptive analysis of HIV risk transmission

Overall, 84.0% (1152/1372) of the study population had unstable aviremia and 45.9% (629) reported inconsistent condom use with at least one negative or unknown HIV status sexual partner (See S2 Table). As a result, 39% (540/1372) were at risk of transmitting HIV (Fig 3). Of these, 26% (144/540) reported two or more partners (versus 16% of patients who were not at risk), 70% reported low ART adherence and 28% had a detectable viral load (versus 54% and 20%, respectively).

The proportions of patients at risk of transmitting HIV varied across HIV services from 25% to 63% (Fig 3). In terms of HSP, the lowest proportion of patients at risk (32%) was observed in HSP1 (HSP2 40%, HSP3 43%, and HSP4 38%).

## Factors associated with HIV transmission risk

Individual and healthcare supply-related factors associated with HIV transmission risk in univariate and multivariable analyses are presented in Table 1. In the final multivariable model adjusted for age, the following individual-related factors were significantly associated with HIV transmission risk: being a woman (adjusted Odds Ratio (aOR) [95% Confidence Interval

**Table 1. Individual and healthcare supply-related factors associated with HIV transmission risk (EVOLCam survey, ANRS 12288).**

| | Total N = 1372 | Univariate analysis N = 1372 | | Multivariable analysis* N = 1367 | |
|---|---|---|---|---|---|
| | n (%) or median [IQR] | OR [95% CI] | p-value | AOR [95% CI] | p-value |
| **Demographic and socioeconomic characteristics** | | | | | |
| Age (in years) | 39 [33–44] | 0.98 [0.97–0.99] | <0.001 | 0.99 [0.98–1.01] | 0.476 |
| Gender | | | | | |
| Men | 447 (32.6%) | 1 | | 1 | |
| Women | 925 (67.4%) | 1.77 [1.39–2.25] | <0.001 | 2.13 [1.60–2.82] | <0.001 |
| Residential setting (MD = 13) | | | | | |
| Urban | 1154 (84.9%) | 1 | | | |
| Rural | 205 (15.1%) | 1.13 [0.81–1.58] | 0.487 | | |
| Marital status (MD = 23) | | | | | |
| Single | 193 (14.3%) | 1 | | | |
| Married (legal or customary) | 539 (40.0%) | 0.71 [0.51–0.99] | 0.049 | | |
| Common-law union | 617 (45.7%) | 1.11 [0.79–1.55] | 0.534 | | |
| Having a main partner | | | | | |
| Yes | 1179 (85.9%) | 1 | | | |
| No | 193 (14.1%) | 1.10 [0.81–1.52] | 0.529 | | |
| Living with main partner (MD = 19) | | | | | |
| Yes | 757 (55.9%) | 1 | | | |
| No | 403 (29.8%) | 1.50 [1.17–1.93] | 0.002 | | |
| Did not have a main partner | 193 (14.3%) | 1.27 [0.92–1.77] | 0.149 | | |
| Number of children | | | | | |
| None | 153 (11.2%) | 1 | | | |
| 1–4 | 950 (69.2%) | 1.11 [0.78–1.59] | 0.559 | | |
| ≥5 | 269 (19.6%) | 1.14 [0.75–1.72] | 0.547 | | |
| Currently desiring or trying to have a child (MD = 6) | | | | | |
| Yes | 780 (57.1%) | 1 | | | |
| No | 586 (42.9%) | 0.85 [0.68–1.06] | 0.161 | | |
| Educational level (MD = 7) | | | | | |
| Lower than secondary school | 1225 (89.7%) | 1 | | | |
| Secondary school and above | 140 (10.3%) | 0.76 [0.52–1.11] | 0.157 | | |
| Household monthly income per adult-equivalent (in USD) | 14.7 (21.8) | 1.00 [0.99–1.00] | 0.829 | | |
| Economic activity | | | | | |
| Yes | 933 (68.0%) | 1 | | 1 | |
| No | 439 (32.0%) | 1.40 [1.10–1.77] | 0.005 | 1.34 [1.05–1.72] | 0.019 |
| **Sexual behaviors and psychosocial characteristics** | | | | | |
| Number of sexual partners (12 months prior to survey) | | | | | |
| <2 | 1097 (80.0%) | 1 | | 1 | |
| ≥2 | 275 (20.1%) | 1.99 [1.52–2.62] | <0.001 | 2.45 [1.83–3.29] | <0.001 |
| Number of sexual partners in lifetime (MD = 1) | | | | | |
| ≤10 sexual partners | 959 (69.9%) | 1 | | | |
| >10 male sexual partners | 108 (7.9%) | 1.26 [0.84–1.90] | 0.264 | | |
| >10 female sexual partners | 202 (14.7%) | 0.85 [0.62–1.17] | 0.331 | | |
| Unknown | 102 (7.4%) | 0.72 [0.46–1.12] | 0.145 | | |
| Transactional sex (12 months prior to survey) | | | | | |
| Yes (bought or sold sex) | 33 (2.4%) | 1 | | | |
| No | 1334 (97.6%) | 0.57 [0.28–1.15] | 0.116 | | |
| Mental quality of life (range 0–100) (per unit) | 45.0 (14.0) | 0.99 [0.98–1.01] | 0.381 | | |

(*Continued*)

**Table 1.** (Continued)

| | Total N = 1372 | Univariate analysis N = 1372 | | Multivariable analysis* N = 1367 | |
|---|---|---|---|---|---|
| | n (%) or median [IQR] | OR [95% CI] | p-value | AOR [95% CI] | p-value |
| HIV-related stigma score (range 0–8) | 0.0 [0–1] | 1.04 [0.98–1.09] | 0.182 | | |
| Binge drinking (MD = 5) | | | | | |
| < once a month | 1266 (92.6%) | 1 | | | |
| At least once a month | 101 (7.4%) | 0.85 [0.55–1.30] | 0.445 | | |
| **Clinical characteristics** | | | | | |
| Time since HIV diagnosis (years) | 4.0 [2–7] | 0.95 [0.92–0.98] | 0.002 | 0.96 [0.92–0.99] | 0.024 |
| Disease symptoms at HIV diagnosis (MD = 2) | | | | | |
| No | 593 (43.3%) | 1 | | 1 | |
| Yes | 777 (56.7%) | 1.24 [0.99–1.55] | 0.062 | 1.38 [1.08–1.75] | 0.011 |
| Time between HIV diagnosis and ART initiation (MD = 3) | | | | | |
| <2 months | 492 (35.9%) | 1 | | 1 | |
| ≥2 months | 753 (55%) | 1.21 [0.95–1.54] | 0.123 | 1.32 [1.02–1.71] | 0.034 |
| NC (not treated) | 124 (9.1%) | 2.20 [1.45–3.31] | <0.001 | 2.28 [1.48–3.49] | <0.001 |
| CD4 count at ART initiation (MD = 105) | | | | | |
| ≥100 cells/mm³ | 866 (68.4%) | 1 | | | |
| <100 cells/mm³ | 277 (21.9%) | 1.01 [0.76–1.34] | 0.934 | | |
| NC (not treated) | 124 (9.8%) | 1.90 [1.28–2.81] | 0.001 | | |
| **Healthcare service characteristics** | | | | | |
| HIV-service profiles | | | | | |
| HSP1 (n = 4) | 222 (16.2%) | 1 | | 1 | |
| HSP2 (n = 5) | 414 (30.2%) | 1.35 [0.81–2.23] | 0.247 | 1.36 [0.82–2.25] | 0.235 |
| HSP3 (n = 6) | 451 (32.9%) | 1.65 [1.02–2.70] | 0.043 | 1.71 [1.05–2.79] | 0.031 |
| HSP4 (n = 4) | 285 (20.8%) | 1.32 [0.77–2.25] | 0.309 | 1.21 [0.71–2.07] | 0.481 |

*AIC of the final selected model = 1746.74; ICC in the final model = 0.019.

Abbreviations: AIC: Akaike's Information Criterion; ICC: intra-class correlation coefficient; IQR: InterQuartile range; ART: Antiretroviral treatment; MD: Missing Data; NC = Not concerned; HSP: HIV-service profiles.

(CI)]: 2.13 [1.60–2.82]), not having an economic activity (aOR [95% CI]: 1.34 [1.05–1.72]), having had at least two sexual partners in the 12 months prior to the survey (aOR [95% CI]: 2.45 [1.83–3.29]), reporting HIV disease symptoms at diagnosis (aOR [95% CI]: 1.38 [1.08–1.75]), initiating ART two months or more after HIV diagnosis (aOR [95% CI]: 1.38 [1.08–1.75]), and not being treated with ART (aOR [95% CI]: 1.38 [1.08–1.75]). Conversely, longer time since HIV diagnosis was associated with a lower risk of transmitting HIV (0.96 [0.92–0.96] per one year increase).

With regard to healthcare supply-related characteristics, only patients followed in HSP3 had a higher risk of transmitting HIV than those in HSP1 (aOR [95% CI]: 1.71 [1.05–2.79], p = 0.031). (HSP2 (aOR [95% CI]: 1.36 [0.82–2.25]; p = 0.235 and HSP4 (aOR [95% CI]: 1.21 [0.71–2.07]; p = 0.481).

## Discussion

Our study highlighted that a high proportion (39%) of PLHIV attending HIV services in the Centre and Littoral regions in Cameroon had a risk of transmitting HIV, defined here as both unstable aviremia and inconsistent condom use with at least one of the two (if more than one partner reported) most recent partners of negative or unknown HIV status in the previous 12

months. In addition, multivariable models indicated that at-risk patients were more likely to experience a sub-optimal HIV care cascade with late HIV diagnosis (when symptomatic), no ART initiation, and late ART initiation (>2 months after HIV diagnosis) all independently associated with HIV transmission risk. The prevalence of HIV transmission risk also varied greatly across HIV services (from 25% to 63%), suggesting that besides individual factors, healthcare supply characteristics may also play a key role in HIV transmission risk, specifically the implementation of task-shifting practices and support activities, as well as ARV availability.

The prevalence of HIV transmission risk in our study (39%) was higher than values found in three previous studies conducted in Sub-Saharan Africa using a relatively similar comprehensive approach which combined viral load with inconsistent condom use to define the risk of HIV transmission (5%, 10% and 27% in ART-treated patients in Uganda [18], Cote d'Ivoire [6], and Cameroon [17], respectively). This may partly be explained by our definition of HIV transmission risk, which differed from these studies in two ways: first, non-ART treated patients were not excluded and were considered to be at risk of transmitting HIV when they did not consistently use condoms with their sexual partners; second, besides viral load, we also took into account ART adherence to measure unstable aviremia, as ART adherence is a key predictor of virological suppression [23, 44]. Furthermore, those studies were conducted in an experimental setting (randomized trials in Uganda [18] and Cameroon [17], and a cohort study in Côte d'Ivoire [6]), whereas ours was conducted in the context of a real-world program.

Our data indicate a high prevalence of both inconsistent condom use and unstable aviremia in HIV-positive patients linked to care in the Cameroonian ART access program. Although three-quarters of the study population had one or more recent negative or unknown HIV status partner(s), only half (54%) reported consistent condom use with that (those) partner(s). In addition, although most patients (82%) were on ART for at least six months, only a minority were highly adherent (26%) and had detectable viral (28%), resulting in a high proportion (84%) of study patients with unstable aviremia status. These findings reveal the huge challenges still facing HIV services in Cameroon to achieve the third '95' UNAIDS target (i.e., VLS in 95% of PLHIV treated with ART by 2025) and to prevent new HIV infections. They also highlight the need to further develop supportive adherence services (e.g., peer-to-peer support, adherence clubs, and short message services) which have been shown to have a positive impact on adherence and viral suppression among PLHIV in LMIC [45].

Our findings also demonstrate the decisive impact of the two first steps of the HIV care cascade (i.e., HIV diagnosis and ART initiation) on the risk of HIV transmission and reflect results from other studies which highlighted that early difficulties encountered by HIV-positive patients in the continuum of care may compromise future treatment success [46–48]. In addition, these two steps are key predictors of VLS [44, 49] which in turn impacts HIV transmission risk. This highlights the relevance of the test-and-treat strategy currently being rolled out in LMIC, including in Cameroon. In particular, the positive impact of this strategy on the third step of the HIV care cascade has recently been highlighted in four randomized population-based trials [50]. Furthermore, several studies have shown that ART initiation, including early initiation through the 'test-and-treat' strategy, leads to a significant decrease in sexual risk behaviors [17, 51–53] thanks to greater opportunities for counselling and psychological support arising out of more frequent contact with HIV services [54]. Such a decrease may also contribute to lower the risk of HIV transmission.

Finally, using detailed information on resources and management characteristics of the 19 HIV services participating in EVOLCam, we identified four HIV service profiles to explain the differences observed in terms of HIV transmission risk. After adjustment for individual

factors, the multivariable analysis showed that patients followed in HSP3 had a greater risk than those followed in HSP1, while no significant differences were found for HSP2 and HSP4. The description of each profile's structural characteristics enabled us to identify healthcare supply-related factors possibly explaining this higher risk in HSP3 than in HSP1.

First, although HIV services in HSP3 had a low number of medical staff (only 1/6 had more than 4 physicians and 2 nurses), task shifting for consultation was rarely practiced. Conversely HSP1 services had slightly more staff (half had more than 4 physicians and all had least 2 nurses) and most practiced task-shifting and educational counselling. Several studies conducted in sub-Saharan Africa, including in Cameroon, have highlighted that task-shifting from physicians to nurses as well as providing educational counselling, brings substantial benefits to the continuum of HIV care [14, 17, 20, 55, 56].

Second, in HSP3, more HIV services reported limited access to CD4 count machines and ARV stock-outs than in HSP1. The detrimental effect of the latter problem on treatment adherence and interruption [57, 58], and in turn on the development of resistances and higher mortality [27, 59], has been well documented in the literature. In addition, limited access to CD4 count machines has been associated with late ART initiation [60] and poorer patient satisfaction with ART delivery [61].

Several study limitations should be acknowledged. First, the EVOLCam survey was conducted in only two (Centre and Littoral) of Cameroon's 10 regions and is therefore not representative of the whole population of PLHIV enrolled in the Cameroonian ART access program. However, these two regions were among those with the highest HIV prevalence rate at the time of the study (estimated at 6.6% in the Centre and 4.9% in the Littoral in 2014 [62]) and include the country's two main cities (Yaoundé and Douala). Accordingly, they were the most populated regions with the largest PLHIV populations [63]. Moreover, we selected a representative sample of existing HIV services in both regions and used a random selection procedure for participant inclusion. We therefore obtained quite a comprehensive picture of the Cameroonian ART access program in two key regions of the country.

Second, adherence and sexual behaviors were assessed using patients' self-reports, which may be affected by social desirability bias. However, this bias was limited by the administration of face-to-face questionnaires by independent interviewers trained in the use of non-judgmental approaches. In addition, we used a validated ART adherence scale [37] which has already been successfully implemented in several studies conducted in Cameroon [20, 64–66]. The availability of data for viral load measurements for patients on ART for at least six months also helped us to confirm associations between VLS and our adherence variable [35].

Third, Cameroon's national ART access program has seen several substantial changes since 2014 (our study period), including the adoption of the test-and-treat strategy in 2016 and the implementation of free HIV care in the public sector in 2019. These two policies have brought about major progress in terms of ART access, with ART coverage standing at 77% in 2020 [67]. However, this rapid and large increase in the number of ART-treated patients constitutes a huge burden on Cameroon's healthcare system, especially in terms of human resources and drug supplies [68]. Recent studies in the country documented important patient-reported barriers to accessing HIV services, including long waiting times, poor patient reception in centers, poor coordination between HIV testing and ART services, long delays before ART imitation [69], and a higher risk of loss-to-follow-up among patients who initiated early ART [70]. In addition, the country's "free access" policy generates a loss of income for healthcare facilities [3, 71] which may negatively affect healthcare quality (e.g., through increased drug stock-outs and reduced staff motivation) when not offset by government subsidizing [72, 73]. Finally, the ongoing COVID-19 pandemic may result in financial resources being diverted, which may further exacerbate human resources shortages and inadequate ART supplies [74]. Recent

literature [75] and the latest estimation of the proportion of PLHIV achieving VLS in Cameroon (approximately 70% in 2020) [67], suggest that the risk of HIV transmission is still high in Cameroon and that the challenges highlighted in our study are still very much a reality.

Despite these limitations, our findings have important implications for HIV healthcare policy.

First, they suggest that the whole HIV continuum of care should be considered to improve early HIV diagnosis, engagement in care and treatment, with a view to increasing the likelihood of achieving viral suppression and limiting HIV transmission risk. It is also essential to identify patients at risk of transmitting HIV at each step of the continuum to provide them timely support, thereby enabling them to cope with their difficulties and better manage their HIV infection. As viral load measurement is rarely available in routine care in sub-Saharan Africa, targeting patients with low adherence and/or risky sexual behaviors may be an effective approach, especially given that low adherence to ART was found to be predictive of risky sexual behaviors during the first year after ART initiation in Cameroon [76]. Second, given the country's severe human resource shortages, task-shifting for HIV consultations from physicians to nurses should be fostered [77]. Finally, the implementation of interventions to improve ART supply chain efficiency and management are urgently required. A previous study in Cameroon recommended implementing a set of harmonization and coordination procedures, as well as continuous quality control monitoring throughout the whole ARV supply chain to ensure permanent availability of these drugs for PLHIV [78]. Previous studies conducted in sub-Saharan Africa also indicated that HIV services which implement interventions focused on anticipating the needs for and purchase of ART (at the logistics and supply chain levels) and which allocate sustainable financial support to competent resources at all levels of the health system, are the more effective [79–82].

## Conclusions

We used a multi-level logistic regression model to disentangle effects of demand-side factors from those of supply-side factors in order to gain a better understanding of the role played by each of these factors on HIV transmission risk in the context of the Cameroonian ART access program. Our findings suggest that early HIV testing and rapid ART initiation may positively influence the engagement of HIV-positive individuals in prevention strategies. They also highlight that HIV transmission risk was higher in HIV services which had limited human resources and did not practice task-shifting, as well as in those with low technical capacity to ensure regular supply of ARV. Adequate resources should be allocated to all HIV services to both relieve healthcare supply-related barriers and enable them to provide adequate support activities to patients throughout the continuum of care, in order to optimize the individual and public health benefits of ART on the reduction of new HIV infections.

## Supporting information

**S1 Table. Characteristics of the 19 participating healthcare services and HIV service profiles (EVOLCam survey, ANRS-12288).** Abbreviations: ARV = Antiretroviral drugs HSP: HIV-service profile.
(DOCX)

**S2 Table. Description of the variables of interest for the description of the HIV transmission risk outcome (n = 1372, EVOLCam survey, ANRS-12288).** Abbreviation: NC: not concerned; MD: missing data; ART: antiretroviral treatment; HSP: HIV-service profile.
(DOCX)

**S1 Fig. Multiple correspondence analysis (MCA) coordinate plots of categories of active variables (characteristics of centers) (EVOLCam survey, ANRS 12288).**
(DOCX)

**S2 Fig. Multiple correspondence analysis (MCA) coordinate plots of centers (as supplementary variable) and clustering: HSP 1 (S1-S4), HSP 2 (S5-S9), HSP 3 (S10-S15), HSP 4 (S16-S19) (EVOLCam survey, ANRS 12288).**
(DOCX)

**S3 Fig. Dendrogram for center clustering (EVOLCam survey, ANRS 12288).**
(DOCX)

## Acknowledgments

We thank all the study participants and the staff at the participating HIV services who agreed to participate in the study. Our thanks also to the French National Agency for Research on HIV/AIDS and viral hepatitis (ANRS) for its financial support and to Jude Sweeney (Milan, Italy) for the English revision and editing of the manuscript.

The EVOLCAM study group: C. Kuaban, L. Vidal (principal investigators); G. Maradan, A. Ambani, O. Ndalle, P. Momo, C. Tong (field coordination team); S. Boyer, L. March, M. Mora, L. Sagaon-Teyssier, M. de Sèze, B. Spire, M. Suzan-Monti (UMR1252 –SESTIM); C. Laurent, F. Liégeois, E. Delaporte, V. Boyer, S. Eymard-Duvernay (TransVIHMI); F. Chabrol, E. Kouakam, O. Ossanga, H. Essama Owona, C. Biloa, M.-T. Mengue (UCAC); E. Mpoudi-Ngolé (CREMER); P.J. Fouda, C. Kouanfack, H. Abessolo, N. Noumssi, M. Defo, H. Meli (Hôpital Central, Yaoundé); Z. Nanga, Y. Perfura, M.Ngo Tonye, O. Kouambo, U. Olinga, E Soh (Hôpital Jamot, Yaoundé); C. Ejangue, E. Njom Nlend, A. Simo Ndongo (Hôpital de la Caisse, Yaoundé); E Abeng Mbozo'o, M. Mpoudi Ngole, N. Manga, C. Danwe, L. Ayangma, B. Taman (Hôpital Militaire, Yaoundé); E.C. Njitoyap Ndam, B. Fangam Molu, J. Meli, H. Hadja, J. Lindou (Hôpital Général, Yaoundé); J.M. Bob Oyono, S. Beke, (Hôpital Djoungolo, Yaoundé); D. Eloundou, G. Touko, (District Hospital, Sa'a); J.J. Ze, M. Fokoua, L.Ngum, C. Ewolo, C.Bondze (District Hospital, Obala); J.D. Ngan Bilong, D. S.Maninzou, A. Nono Toche (Hôpital St Luc, Mbalmayo); M.Tsoungi Akoa, P. Ateba, S. Abia (District Hospital, Mbalmayo); A. Guterrez, R. Garcia, P. Thumerel (Catholic Health Centre, Bikop); E. Belley Priso, Y Mapoure, A. Malongue, A.P. Meledie Ndjong, B. Mbatchou, J. Hachu, S. Ngwane (Hôpital Général, Douala); J. Dissongo, M. Mbangue, Ida Penda, H. Mossi, G. Tchatchoua, Yoyo Ngongang, C.Nouboue, I. Wandji, L. Ndalle, J. Djene (Hôpital Laquintinie, Douala); M.J. Gomez, A. Mafuta, M. Mgantcha (Catholic Hospital St Albert Legrand, Douala); E.H. Moby, M.C. Kuitcheu, A.L. Mawe, Ngam Engonwei (District Hospital, Bonassama); L.J. Bitang, M. Ndam, R.B.Pallawo, Issiakou Adamou, G.Temgoua (District Hospital, Deido); C.Ndjie Essaga, C. Tchimou, A. Yeffou, I. Ngo, H. Fokam, H. Nyemb (District Hospital, Nylon); L.R. Njock, S. Omgnesseck, E. Kamto, B. Takou (District Hospital, Edea); L.J-G Buffeteau, F. Ndoumbe, J-D Noah, I. Seyep (Hôpital St Jean de Malte, Njombe).

## Author Contributions

**Conceptualization:** Pierre-julien Coulaud, Luis Sagaon-Teyssier, Christian Laurent, Bruno Spire, Laurent Vidal, Christopher Kuaban, Sylvie Boyer.

**Formal analysis:** Pierre-julien Coulaud, Abdourahmane Sow, Luis Sagaon-Teyssier, Khadim Ndiaye, Sylvie Boyer.

**Investigation:** Gwenaëlle Maradan, Christian Laurent, Laurent Vidal, Christopher Kuaban, Sylvie Boyer.

**Methodology:** Luis Sagaon-Teyssier, Sylvie Boyer.

**Project administration:** Gwenaëlle Maradan.

**Supervision:** Pierre-julien Coulaud, Luis Sagaon-Teyssier, Bruno Spire, Sylvie Boyer.

**Writing – original draft:** Pierre-julien Coulaud, Sylvie Boyer.

**Writing – review & editing:** Pierre-julien Coulaud, Luis Sagaon-Teyssier, Christian Laurent, Bruno Spire, Sylvie Boyer.

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
