## [Decision Letter · Decision Letter 0]

21 Sep 2021

PONE-D-21-22603Individual and healthcare supply-related HIV transmission factors in HIV-positive patients enrolled in the Cameroonian antiretroviral treatment access program (ANRS-12288 EVOLCam survey)PLOS ONE

Dear Dr. Coulaud,

Thank you for submitting your manuscript to PLOS ONE. After careful consideration, we feel that it has merit but does not fully meet PLOS ONE’s publication criteria as it currently stands. Therefore, we invite you to submit a revised version of the manuscript that addresses the points raised during the review process Thank you for submitting your manuscript. We have reviewed it and hereby make the below recceommendations. 

Seeing that this data is almost 7 years old, kindly highlight the validity of the results in 2021, because some progresses have been made so far by the Government of Cameroon. Additionally, please update the information shared to reflect recent surveys in Cameroon. Please also address the issues raised by the various reviewers- as attached.

We look forward to receiving your revised manuscript.

Kind regards,

Judith Kose, M.D.

Academic Editor

PLOS ONE

Journal Requirements:

2. Thank you for submitting the above manuscript to PLOS ONE. During our internal evaluation of the manuscript, we found significant text overlap between your submission and the following previously published works, some of which you are an author.

https://academic.oup.com/heapol/article/36/2/137/6050511

Please revise the manuscript to rephrase the duplicated text, cite your sources, and provide details as to how the current manuscript advances on previous work. Please note that further consideration is dependent on the submission of a manuscript that addresses these concerns about the overlap in text with published work.

3. Please include additional information regarding the survey or questionnaire used in the study and ensure that you have provided sufficient details that others could replicate the analyses. For instance, if you developed a questionnaire as part of this study and it is not under a copyright more restrictive than CC-BY, please include a copy, in both the original language and English, as Supporting Information

4. In ethics statement in the manuscript and in the online submission form, please provide additional information about the patient records/samples used in your retrospective study. Specifically, please ensure that you have discussed whether all data/samples were fully anonymized before you accessed them

Reviewers' comments:

Reviewer's Responses to Questions

**Comments to the Author**

1. Is the manuscript technically sound, and do the data support the conclusions?

Reviewer #1: Yes

Reviewer #2: Yes

Reviewer #3: Yes

2. Has the statistical analysis been performed appropriately and rigorously? 

Reviewer #1: Yes

Reviewer #2: Yes

Reviewer #3: Yes

3. Have the authors made all data underlying the findings in their manuscript fully available?

Reviewer #1: Yes

Reviewer #2: Yes

Reviewer #3: Yes

4. Is the manuscript presented in an intelligible fashion and written in standard English?

Reviewer #1: Yes

Reviewer #2: Yes

Reviewer #3: Yes

5. Review Comments to the Author

Reviewer #1: 1. Since the regions of Littoral and Centre are not representative of the entire country, please insert Littoral and Centre Regions in the title of the manuscript.

2. Line 6: These are objectives, not aims

3. Please rewrite the introduction to update the situation in Cameroon (with HIV figures), and show the link of this work with new HIV national strategic plan.

4. The data were collected between April and December 2014, now 7 years back. Please discuss the validity of the results in 2021, because some progresses have been made so far by the Government of Cameroon.

5. Briefly describe the EVOLCam survey.

6. In line 428, the authors stated the “Due to French law there are restrictions on publicly sharing the data of this study. French law requires that everyone who wishes to access cohorts data or clinical study data on humans must ask the French data protection authority”. However, the study was conducted in Cameroon and received ethical approval from Ethical Board of Cameroon, not French one. How was the data sharing agreement with Cameroon partners? Please clarify this issue.

7. Are there any standard definitions of “the risk of transmitting HIV” and “Unstable aviremia”? if so give references otherwise justify.

8. Please note that the Ministry of Public Health gives the Admistrative Autorisation while the ethical approval is given by the “Comité National d’Ethique de la Recherche pour la Santé Humaine (CNERSH)”

9. The study was in 19 HIV services in Cameroon’s Center and Littoral, please give the total number of HIV services in these two regions.

10. The justification of the choice of the two regions is not clear as the South Region has the highest HIV prevalence rate.

11. 1 372 should clearly appear in Figure 2, this is because, 10 the number missing data on viral load and/or ART adherence is not included there.

12. Please insert the goodness of fit of the final selected multilevel model.

13. Interpret the random component summarized by inter-class variance.

14. Are there clear reasons why the final multivariable model was adjusted for age only, not for both age and sex as mentioned in line 275?

15. In the Abstract, add p-values to CI of Adjusted ORs, and IQR to the median of age.

Minor

16. Affiliation 4 is « Faculté de Médecine et de sciences biomédicales, Université de Yaoundé 1, Yaoundé, Cameroun »

17. Rewrite line 101

Reviewer #2: This paper is easy to read and relevant within the epidemiological context of eradicating HIV pandemic in Cameroon

It should be good for the authors to update some informations in the study settings on the prevalence of HIV in Cameroon line 94, in accordance with latest DHS 2018.

In addition the context , regardless of TASP, should outline the notion of test and treat, and the model of service for ART delivery can be described following line 102...for instance

The surprising fact in table 1 is the rate of patient not yet treated, assuming Cameroon at this stage was yet in the test and treat approach. Can the authors have a clear idea of why those patients were not under treatment as this can appear to be a confounding risk.

Line 299-301 Explain the need of test and treat and not delaying ART

Line 324-326-emphazise and discuss supportive adherence counseling after ART initiation

Line 328-331-Discuss test and treat strategy to address HIV care and treatment cascade

Final discussion and recommendations can be more focused on the importance of test and treat, task shifting and ARV stocks management to prevent ARV stock outs

Reviewer #3: In terms of content/form, the document seems to me to be well written. The subject addresses an issue never before addressed in Cameroon to the best of my knowledge and worthy of interest for improving the national response against HIV in Cameroon. The ethical aspects are not well described, but all the required authorizations have been obtained. The methodology is simple and clear and the conclusions are linked to it.

6. PLOS authors have the option to publish the peer review history of their article (what does this mean?). If published, this will include your full peer review and any attached files.

Reviewer #1: **Yes: **Georges Nguefack-Tsague

Reviewer #2: **Yes: **Anne Esther NJOM NLEND

Reviewer #3: **Yes: **Serge Clotaire BILLONG

---

## [Author Response · Author response to Decision Letter 0]

14 Jan 2022

Dr. Pierre-julien Coulaud, PhD

Aix Marseille University, INSERM, IRD, SESSTIM, Sciences Economiques & Sociales de la Santé & Traitement de l’Information Médicale

Faculté de Médecine Timone, 27 Boulevard Jean Moulin – 13005 Marseille, France

Emails: pierre-julien.coulaud@inserm.fr / pierre-julien.coulaud@bccsu.ubc.ca

Marseille, 10th December, 2021

Manuscript Number PONE-D-21-22603

Title: " Individual and healthcare supply-related HIV transmission factors in HIV-positive patients enrolled in the Cameroonian antiretroviral treatment access program (ANRS-12288 EVOLCam survey)"

Dear Editor, 

Thank you very much for the evaluation of our revised manuscript for publication in PLOS ONE and for giving us the opportunity to submit a new revised version. 

We carefully considered and replied to all the editor and reviewers’ comments (see below). In addition, we made some minor corrections in the text (for English corrections and editing). All revisions made in the new version of the manuscript are highlighted in yellow. 

More specifically, we addressed the following three major points raised by the editor and reviewers. First, we revised our rational in the Introduction to better explain how our manuscript advances on our previous work (Coulaud et al., 2021). While this previous work was focused on understanding the role of healthcare supply-related factors on ART initiation delay (which correspond to the second UNAIDS target), our present manuscript investigates the association between these structural factors and HIV risk transmission, corresponding to the third UNAIDS target. We did not mention this publication in our first submission to your journal because this previous article was not published when we finalized and first submitted this manuscript in December 2020 (it was, at this time, reviewed by another journal). As some parts of our manuscript was overlapping with this previous work, we carefully revised this new version to remove the duplicated text and our references were updated accordingly.

Second, we provided additional information regarding the survey and the ethical statement. Given that we used a similar approach for defining our HIV services profiles that in our previous publication (i.e., cluster analysis), we reorganized our Methods and Results sections to limit similar extracts from our previous work and summarize the main findings. Regarding the access to the study questionnaire, we specified that they have been previously published as supplement materials in Fiorentino et al., 2021. We also revised our ethical statement to clarify that each participant was assigned with a patient identification number that was used to anonymize all data collection tools, including the datasets for conducting this analysis.

Third, as suggested by reviewers, the discussion and limits section of our manuscript were modified to highlight the implications of the test-and-treat strategy on the HIV treatment-related outcomes (especially on access to treatment, viral suppression and sexual risk behaviours). In addition, as our data were collected in 2014, we better discussed the recent evolutions of the Cameroonian ART Program as well as the relevance of our findings and public health policy recommendations in the current context. 

Finally, as suggested by the first reviewer, we also modified the title to specify the study area as follows: “Individual and healthcare supply-related HIV transmission factors in HIV-positive patients enrolled in the antiretroviral treatment access program in the Centre and Littoral regions in Cameroon (ANRS-12288 EVOLCam survey)”.

We hope that this new version will be suitable for publication in PLOS ONE and remain at your disposal for any request you may have. 

Yours faithfully, 

Pierre-julien Coulaud, on behalf of the co-authors 

ANSWERS TO REVIEWERS' COMMENTS

Reviewer #1.

1. Since the regions of Littoral and Centre are not representative of the entire country, please insert Littoral and Centre Regions in the title of the manuscript.

As suggested, we modified the title by inserting “Littoral and Centre regions” as follows: 

“Individual and healthcare supply-related HIV transmission factors in HIV-positive patients enrolled in the antiretroviral treatment access program in the Centre and Littoral regions in Cameroon (ANRS-12288 EVOLCam survey)”

2. Line 6: These are objectives, not aims

As suggested, we revised this sentence (line 71-76, page 4) as follows: 

“In a previous study conducted in Cameroon, we showed that HIV service profiles, built using a cluster analysis of a wide range of healthcare supply-related characteristics, had different performances in terms of time to ART initiation [1]. In the present study, we used a similar approach to provide a better understanding of the role of supply-related factors, beside individual factors, on achieving the third UNAIDS target in Cameroon.”

3. Please rewrite the introduction to update the situation in Cameroon (with HIV figures), and show the link of this work with new HIV national strategic plan.

We revised the Introduction section to update the last estimations and references available regarding the HIV epidemic and the 2025 UNAIDS testing and treatment targets. A detailed study context is also presented in the sub-section “Study setting” at the beginning of the Methods section (p. 5/6). This one has been revised to include the most recent HIV prevalence rate (as required by Reviewer 2; see comment #18 below). 

As suggested, we also clarified in the Introduction how our study objectives linked to the national HIV strategic plan priorities in Cameroon and provided in the “Study setting subsection” additional information on the priority strategies and interventions promoted by the national health authorities within this plan. 

Introduction (see p.4/5, lines 71-79): 

“In a previous study conducted in Cameroon, we showed that HIV service profiles, built using a cluster analysis of a wide range of healthcare supply-related characteristics, had different performances in terms of time to ART initiation [1]. In the present study, we used a similar approach to provide a better understanding of the role of supply-related factors, beside individual factors, on achieving the third UNAIDS target in Cameroon. This study also provided the opportunity to highlight challenges related to the implementation of the 2018-2022 Cameroonian National Strategic Plan for HIV/AIDS and STIs which aims to reduce the number of new HIV infections by 60% and achieve VLS in 92% of PLHIV on ART in Cameroon by 2022 [2].”

Study setting section, Methods (see p.5/6, lines 95-104):

“The Cameroon national authorities decided to provide ART for free in 2007 and to remove all user fees for HIV care in 2019 [3]. Thanks to this continuously developing HIV strategy, the total number of PLHIV on ART increased from 17,940 in 2005 to 145,038 in 2014 and 367,871 in 2020 [4,5].

In its latest National Strategic Plan for HIV/AIDS and STIs for the period 2018-2022 [2], Cameroonian health authorities promote a set of priority strategies and interventions including i) strengthening HIV prevention, ii) scaling-up HIV testing and ART access through the decentralization of HIV care, task shifting, and the involvement of community-based organizations, iii) ensuring the permanent availability of laboratory equipment, antiretroviral drugs and medicines of opportunistic infections.” 

4. The data were collected between April and December 2014, now 7 years back. Please discuss the validity of the results in 2021, because some progresses have been made so far by the Government of Cameroon.

We acknowledge that our study is dated and that some progresses have been made so far by the Government of Cameroon, including the adoption of the test-and-treat strategy and the gratuity of all HIV care. However, our policy recommendations seems to be still relevant in the current Cameroonian context for the following two main reasons: first, the financial resources of the Cameroonian ART programme have not sufficiently increased between 2014 and 2020 [6] to fill the structural gaps highlighted in our manuscript. Second, the test-and-treat strategy and the gratuity of HIV care adopted by the Government led to a large increase in the numbers of PLHIV initiating ART. The consequence is a greater burden on the health system, especially on human resources which have less time to devote to patients [7]. HIV services are thus expected to initiate ART quickly in all newly diagnosed HIV-positive patients while continuing to provide clinical follow-up and psychosocial support to a growing number of patients already being treated. 

These challenges have been recently documented in HIV services in Cameroon. In 2018, a qualitative study on patients’ satisfaction with ART services indicated that healthcare providers were facing a rapid increase in the volume of patients on ART, especially an increased number of asymptomatic patients who were more hesitant toward the benefits of ART [8]. During the interviews, PLHIV also expressed barriers to engage in HIV services such as experiencing long waiting time, poor reception and attitudes of some healthcare providers (e.g., inadequate counselling and rushing patients to initiate ART due to initial misunderstanding of the test and treat principle), poor coordination between HIV testing and treatment services, and lack of flexibility of the drug delivery system to patients’ specific needs [8]. A retrospective study also reported mixed results regarding the impact of the test and treat policy on HIV care continuum outcomes. Although higher ART uptake, and earlier ART initiation were found among patients who were enrolled after the implementation of the test and treat policy, lower ART retention was reported in this group compared to those who initiated before the implementation of this strategy [9].

While major progress have been made to improve access to ART among PLHIV in Cameroon (i.e., ART coverage of 77% in 2020) [10], recent data [11,12] showed that approximately 70% of PLHIV were virally suppressed in 2020, suggesting that the risk of HIV transmission is still high in Cameroon and that the challenges highlighted in our study are still very much a reality. 

We therefore revised the Limitation section of the Discussion (see p. 19/20, lines 372-389) to better discuss the recent evolutions in the Cameroonian ART programme and the validity of our results in this context, as follows: 

“Third, Cameroon’s national ART access program has seen several substantial changes since 2014 (our study period), including the adoption of the test-and-treat strategy in 2016 and the implementation of free HIV care in the public sector in 2019. These two policies have brought about major progress in terms of ART access, with ART coverage standing at 77% in 2020 [12]. However, this rapid and large increase in the number of ART-treated patients constitutes a huge burden on Cameroon’s healthcare system, especially in terms of human resources and drug supplies [13]. Recent studies in the country documented important patient-reported barriers to accessing HIV services, including long waiting times, poor patient reception in centers, poor coordination between HIV testing and ART services, long delays before ART imitation [8], and a higher risk of loss-to-follow-up among patients who initiated early ART [9]. In addition, the country’s “free access” policy generates a loss of income for healthcare facilities [6,14] which may negatively affect healthcare quality (e.g., through increased drug stock-outs and reduced staff motivation) when not offset by government subsidizing [7,15]. Finally, the ongoing COVID-19 pandemic may result in financial resources being diverted, which may further exacerbate human resources shortages and inadequate ART supplies [10]. Recent literature [11] and the latest estimation of the proportion of PLHIV achieving VLS in Cameroon (approximately 70% in 2020), [12]) suggest that the risk of HIV transmission is still high in Cameroon and that the challenges highlighted in our study are still very much a reality.” 

5. Briefly describe the EVOLCam survey.

As suggested, we described in greater details how and which data were collected in the EVOLCam survey in the sub-section “Study design and data collection” (see p. 6/7):

“We used data from the cross-sectional survey EVOLCam (ANRS 12288) which was conducted in 19 HIV services in Cameroon’s Centre (n=11) and Littoral regions (n=8) between April and December 2014 to study evolutions in the national ART program through a comparison with the 2006-2007 ANRS-12116 EVAL survey [16,17]. The EVOLCam (ANRS 12288) study protocol is described in detail elsewhere [18,19]. 

Briefly, eligible PLWH (≥21 years old and HIV diagnosed >3 months) attending an outpatient consultation in one of the 19 participating HIV services were randomly selected and informed about the study. Patients willing to participate provided written informed consent before data collection. First, a standardized medical questionnaire was completed during the consultation by healthcare providers. The following clinical data were obtained from patient examinations and retrospective medical files: dates of HIV diagnosis and ART initiation, WHO clinical stage of HIV infection at ART initiation and at the time of the study, CD4 count at ART initiation, drug regimen at the time of the study, body mass index and any history of tuberculosis and hepatitis B co-infection and related diagnosis date. Second, patients answered a face-to-face questionnaire administered in a private room by trained independent interviewers which collected data on demographic, socioeconomic, behavioral, psychosocial and domestic information. More specifically, a series of questions were asked on adherence to ART, perceived health and HIV-related stigma as well as alcohol consumption and sexual behaviors during the 12 months prior to the study (number of sexual partners, experience of transactional sex, frequency of sexual relationships, HIV status and condom use with the two most recent partners). The questionnaire is available as supplementary material in Fiorentino et al., 2021 [20]. Third, a blood sample was taken to measure HIV viral load (only for patients ART treated >6 months) and CD4 cell count. All blood samples were analyzed by a reference HIV laboratory in Yaoundé. 

Finally, detailed data on the characteristics of the participating 19 HIV services were collected through interviews with hospital staff, in situ observations, and cross-validation with data recorded in HIV service activity reports. Specifically, the information obtained included: i) hospital’s general characteristics (location, opening date, legal status, number of beds), ii) human resources working in the HIV service (number and qualifications), iii) activity (number of ART-treated patients and available services including educational, nutritional and financial support, HIV community-based organization involvement), iv) HIV service organization (separate ARV storage, stock management and task-shifting for clinical consultations of ART-treated patients and/or ARV prescription renewals), v) technical resources (functional medical imaging equipment, CD4 count machine, ARV stock-outs for at least one of the three most prescribed ART regimens [1].”

6. In line 428, the authors stated the “Due to French law there are restrictions on publicly sharing the data of this study. French law requires that everyone who wishes to access cohorts data or clinical study data on humans must ask the French data protection authority”. However, the study was conducted in Cameroon and received ethical approval from Ethical Board of Cameroon, not French one. How was the data sharing agreement with Cameroon partners? Please clarify this issue.

Thank you for your comment. The data sharing agreement specify that the EVOLCam survey data was available for analysis to the whole research team (i.e. co-investigators, researchers and trainees from all partners in both countries, Cameroon and France). Furthermore, after having checked the issue of data access for people / institute(s) who are not member of the research team, we confirmed that there is no need to receive previous approval from the CNIL in France if data are fully anonymized. We therefore modified the “Availability of data and materials statement” (see p. 23) as follows: 

“Fully anonymized data are available on request made to the study investigators (Laurent Vidal at Laurent.vidal@ird.fr and Christopher Kuaban at ckuaban@yahoo.fr). 

7. Are there any standard definitions of “the risk of transmitting HIV” and “Unstable aviremia”? if so give references otherwise justify.

Thank you for your comment. As mentioned in the Introduction section (see lines 59-63), various definitions were used in the existing literature to define HIV transmission risk. It is also important to note that this definition is constantly evolving with the progress in HIV care, the development of new prevention strategies (e.g., PrEP), and the availability of individual HIV clinical characteristics (e.g., viral load). 

Most of previous studies were focused on behavioural characteristics (e.g., having condomless sex with a partner of unknown HIV status) [21–23] as information on viral load was more difficult to collect. In order to take into account the fact that multiple factors may influence the potential risk of transmission, our definition of HIV transmission risk was based on both behavioural characteristics (i.e., number of partners, condom use, knowledge of partner’s HIV status, adherence to ART) and biomedical factors (i.e., being on ART, viral load) [20,24–26]. This comprehensive approach of the risk of transmitting HIV allowed us to provide a more accurate estimation of the prevalence of HIV transmission risk in our study sample.

A similar approach was used to define unstable aviremia. Previous studies often used the number of viral load copies (undetectable versus detectable) as the main indicator to consider a participant with unstable aviremia [24,25]. However, it’s well known in the HIV literature that the level of adherence to ART play a critical role to help maintain a low viral load and be undetectable [27]. To capture this important aspect that reflects the engagement of PLHIV in HIV care, we considered as “unstable aviremia”: i) participants on treatment with a detectable viral load and ii) participants on treatment with an undetectable viral load but non-adherent to ART. 

For both definitions, we included references in this new version. We also specified our comprehensive approach in great details to highlight the importance of combining both behavioral and biomedical factors. Below are the changes made in the “Outcome” subsection in the Methods section (p. 8):

 “The study outcome was a binary variable describing the risk of transmitting HIV (yes versus no). No standard method exists to define the risk of HIV transmission. Accordingly, in order to define the outcome, we used the literature to develop a comprehensive approach which included both biomedical (i.e., unstable aviremia) and behavioral factors (i.e., inconsistent condom use with negative or unknown HIV status partner(s)) [20,24–26]. More specifically, we defined the risk of transmitting HIV as a combination of both unstable aviremia and reporting inconsistent condom use either with the most recent (i.e., in the previous 12 months) sexual partner (if only one partner declared), or with at least one of the two most recent sexual partners (if more than one partner declared) of negative or unknown HIV status. 

Unstable aviremia was defined as not currently being treated or on treatment for less than six months or on treatment for more than six months but with a detectable viral and/or poor adherence to ART [20]. The latter was defined as taking <80% of the prescribed drug doses or reporting treatment interruptions for at least two consecutive days in the four weeks prior to the survey [28]. Participants on treatment for more than six months with an undetectable viral load who were highly adherent to ART (defined as taking >80% of the prescribed drug doses in the four weeks prior to the survey) were considered to have stable aviremia.”

8. Please note that the Ministry of Public Health gives the Admistrative Autorisation while the ethical approval is given by the “Comité National d’Ethique de la Recherche pour la Santé Humaine (CNERSH)”

Thank you for this comment. We revised our ethical statement (see p. 23) accordingly:

“This study was conducted in compliance with international and national regulations on ethics and research on people. It received administrative authorisation from the Ministry of Public Health in Cameroon and was approved by the Cameroonian National Ethics Committee (approval reference: 2013/08/349/L/CNERSH/SP). All participants were informed about the study’s objectives and its modalities and all provided written consent to participate. All individual data collected in the research were anonymized using a patient identification number; only this number was reported in the data collection tools and the databases used for analyses.”

9. The study was in 19 HIV services in Cameroon’s Center and Littoral, please give the total number of HIV services in these two regions.

As suggested, we clarified the number of HIV services in each region (see the “Study design and data collection” subsection, p.6 lines 107-110) as follows: 

“We used data from the cross-sectional survey EVOLCam (ANRS 12288) which was conducted in 19 HIV services in Cameroon’s Center (n=11) and Littoral regions (n=8) between April and December 2014 to study evolutions in the national ART program through a comparison with the 2006-2007 ANRS-12116 EVAL survey [16,17].”

10. The justification of the choice of the two regions is not clear as the South Region has the highest HIV prevalence rate.

There are two main reasons of the choice of conducting the EVOLCam survey in the Centre and Littoral regions of Cameroon. First, those two regions were among those with the highest HIV prevalence rate at the time of the study (estimated at 6.6% in the Centre and 4.9% in the Littoral in 2014 [29]) and include the country’s two main cities (Yaoundé and Douala) which both also have a prevalence rate higher than the mean prevalence rate of the country. Second, these two regions had a relatively high number of HIV services (respectively 38 in the Centre region and 16 in the Littoral region), following the largest PLHIV populations in Cameroon at the time of the survey [30]. The justification of the choices of the two regions selected for the study is now provided in the limitation subsection in the Discussion (see p.19, lines 355-364) as follows:

“First, the EVOLCam survey was conducted in only two (Centre and Littoral) of Cameroon’s 10 regions and is therefore not representative of the whole population of PLHIV enrolled in the Cameroonian ART access program. However, these two regions were among those with the highest HIV prevalence rate at the time of the study (estimated at 6.6% in the Centre and 4.9% in the Littoral in 2014 [29]) and include the country’s two main cities (Yaoundé and Douala). Accordingly, they were the most populated regions with the largest PLHIV populations [30]. Moreover, we selected a representative sample of existing HIV services in both regions and used a random selection procedure for participant inclusion. We therefore obtained quite a comprehensive picture of the Cameroonian ART access program in two key regions of the country.”

11. 1372 should clearly appear in Figure 2, this is because, 10 the number missing data on viral load and/or ART adherence is not included there.

As suggested, we specified the total number of patients included in our study analysis and we simplified the presentation of the number of patients with missing data on viral load and/or ART adherence in the revised Figure 2.

12. Please insert the goodness of fit of the final selected multilevel model.

The goodness of fit of our final multilevel was assessed using Akaike’s Information Criterion (AIC). We provided this information in the sub-section “Statistical Analysis” of the Methods (see p.11, line 231) as well as in the notes of the Table 1 (see p.15).

Sub-section “Statistical analysis”, Methods: 

“Model fit was assessed using Akaike’s Information Criterion (AIC)”

Table 1: 

“AIC of the final selected model: 1746.74”.

13. Interpret the random component summarized by inter-class variance.

As suggested, we provided in the sub-section “Statistical Analysis” of the new version of the manuscript further details regarding the interpretation of the inter-class variance. We also computed and interpreted the intraclass correlation coefficient (see p.11, lines 221-227), as follows: 

“Initially, we estimated the empty model (without any explanatory variables) to provide an estimation of the inter-class variance, which was small but significantly different from 0 ( = 0.09; p=0.032), confirming the relevance of using a multilevel model. We also computed the estimated intra-class correlation coefficient (ICC), which represents the proportion of the inter-class variance compared to the total variance (i.e., inter- and intra-class variance). It was estimated at 0.027 indicating that 2.7% of the outcome’s variance was due to differences between HIV services.”

14. Are there clear reasons why the final multivariable model was adjusted for age only, not for both age and sex as mentioned in line 275?

Our final multivariable model was adjusted for both age and gender. As described in the sub-section “Statistical Analysis” (page p.11, line 228), only significant individual variables with a p-value <0.05 were retained in our final model, which was the case for gender (p<0.001) but not for age (p>0.05). Given that age is a key demographic characteristic, we forced this variable in the model. We therefore specified in the text (lines 266-267) that our final model was adjusted by age but gender is also included in the final model as it is significantly associated with the HIV transmission risk.

15. In the Abstract, add p-values to CI of Adjusted ORs, and IQR to the median of age.

The Results section of the Abstract (see p. 1/2) has been revised as suggested: 

“Results: Of the 1372 patients (women 67%, median age [Interquartile]: 39 [33-44] years) reporting sexual activity in the previous 12 months, 39% [min-max across HIV services: 25%-63%] were at risk of transmitting HIV. The final model showed that being a woman (adjusted Odd Ratio [95% Confidence Interval], p-value: 2.13 [1.60-2.82], p<0.001), not having an economic activity (1.34 [1.05-1.72], p=0.019), having at least two sexual partners (2.45 [1.83-3.29], p<0.001), reporting disease symptoms at HIV diagnosis (1.38 [1.08-1.75], p=0.011), delayed ART initiation (1.32 [1.02-1.71], p=0.034) and not being ART treated (2.28 [1.48-3.49], p<0.001) were all associated with HIV transmission risk. Conversely, longer time since HIV diagnosis was associated with a lower risk of transmitting HIV (0.96 [0.92-0.99] per one-year increase, p=0.024). Patients followed in the third profile had a higher risk of transmitting HIV (1.71 [1.05-2.79], p=0.031) than those in the first profile.” 

16. Affiliation 4 is « Faculté de Médecine et de sciences biomédicales, Université de Yaoundé 1, Yaoundé, Cameroun »

This error has been corrected in the revised version (see Title page): 

“Faculté de Médecine et de sciences biomédicales, Université de Yaoundé 1, Yaoundé, Cameroun”

17. Rewrite line 101

As suggested, we rewrote this sentence (see p.5) as follows:

“The Cameroon national authorities decided to provide ART for free in 2007 and to remove all user fees for HIV care in 2019 [3].”

Reviewer #2.

18. This paper is easy to read and relevant within the epidemiological context of eradicating HIV pandemic in Cameroon. It should be good for the authors to update some information in the study settings on the prevalence of HIV in Cameroon line 94, in accordance with latest DHS 2018.

As suggested, we updated the estimation of HIV prevalence rate according to the 2018 Demographic and Health Survey conducted in Cameroon and provided additional estimations on the HIV prevalence according some key characteristics (i.e., region of residence, gender, area of residence: urban vs. rural) (see the subsection “Study setting” in the Methods, p.5, lines 87-91): 

“Cameroon is an LMIC in Central Africa affected by a generalized HIV epidemic with a mean estimated prevalence rate in 2018 of 2.7% in adults (aged 15-49 years), and large disparities according to gender, region and urban area [31]. The highest prevalence rates are observed in women (3.4%), in urban areas (3.9%), and in the South (5.8%), East (5.6%), Adamaoua (4.1%), North-West (4.0%) and Centre (3.5%) [31] regions.” 

19. In addition the context, regardless of TASP, should outline the notion of test and treat, and the model of service for ART delivery can be described following line 102...for instance

As suggested, we revised the Introduction (see p.3) to highlight the importance of the test-and-treat strategy as a key component of the U=U movement to reduce HIV infections and discussed the implications of such strategy on HIV risk transmission and quality of HIV services in the Discussion section (see p. 18-21).

Introduction (see p.3, lines 40-46):

“One key biomedical intervention is early antiretroviral treatment (ART) initiation which has been shown to dramatically reduce HIV-related mortality and morbidity as well as HIV transmission risk [25,32–34]. The beneficial effect of early ART on viral load suppression (VLS) led to the establishment of the U=U (“undetectable equals untransmittable”) movement, which is widely recognized for its importance in controlling the HIV epidemic [27]. This treatment as prevention approach evolved into the test-and-tread strategy, that is to say ART initiation immediately after HIV diagnosis, irrespective of CD4 count [35].”

Discussion section:

See p.18/19, lines 324-332:

“This highlights the relevance of the test-and-treat strategy currently being rolled out in LMIC, including in Cameroon. In particular, the positive impact of this strategy on the third step of the HIV care cascade has recently been highlighted in four randomized population-based trials [36]. Furthermore, several studies have shown that ART initiation, including early initiation through the ‘test-and-treat’ strategy, leads to a significant decrease in sexual risk behaviors [24,37–39] thanks to greater opportunities for counselling and psychological support arising out of more frequent contact with HIV services [40]. Such a decrease may also contribute to lower the risk of HIV transmission.”

See p.20/21, lines 372-389:

“Third, Cameroon’s national ART access program has seen several substantial changes since 2014 (our study period), including the adoption of the test-and-treat strategy in 2016 and the implementation of free HIV care in the public sector in 2019. These two policies have brought about major progress in terms of ART access, with ART coverage standing at 77% in 2020 [12]. However, this rapid and large increase in the number of ART-treated patients constitutes a huge burden on Cameroon’s healthcare system, especially in terms of human resources and drug supplies [13]. Recent studies in the country documented important patient-reported barriers to accessing HIV services, including long waiting times, poor patient reception in centers, poor coordination between HIV testing and ART services, long delays before ART imitation [8], and a higher risk of loss-to-follow-up among patients who initiated early ART [9]. In addition, the country’s “free access” policy generates a loss of income for healthcare facilities [6,14] which may negatively affect healthcare quality (e.g., through increased drug stock-outs and reduced staff motivation) when not offset by government subsidizing [7,15]. Finally, the ongoing COVID-19 pandemic may result in financial resources being diverted, which may further exacerbate human resources shortages and inadequate ART supplies [10]. Recent literature [11] and the latest estimation of the proportion of PLHIV achieving VLS in Cameroon (approximately 70% in 2020), [12]) suggest that the risk of HIV transmission is still high in Cameroon and that the challenges highlighted in our study are still very much a reality.” 

20. The surprising fact in table 1 is the rate of patient not yet treated, assuming Cameroon at this stage was yet in the test and treat approach. Can the authors have a clear idea of why those patients were not under treatment as this can appear to be a confounding risk.

At the time of the EVOLCam survey (i.e., April-December 2014), the test-and-treat strategy was not yet implemented in Cameroon (this strategy was adopted in June 16 [41]). Therefore, a subset of participants (n=124, 9% of our sample) were not treated because they may not be eligible for ART initiation at the time of the survey. Those patients were considered at risk of transmitting HIV if they reported inconsistent condom use with at least one partner of negative or unknown HIV status. In addition, specific sub-categories for patients not treated were created for the “time between HIV diagnosis and ART initiation” and “CD4 count at ART initiation” variables in our univariate and multivariable analysis (see Table 1, p.15). We specified in the Discussion section that the “test-and-treat strategy” was adopted in 2016 and discuss its potential impact on the study findings (see p.18-21). See also our responses to the comments #4 and #19 of the reviewer #1.

21. Line 299-301 Explain the need of test and treat and not delaying ART

As suggested, the importance of the test and treat strategy to improve the HIV care cascade has been emphasized in the Discussion, as follows (see p.18/19, lines 324-332): 

“This highlights the relevance of the test-and-treat strategy currently being rolled out in LMIC, including in Cameroon. In particular, the positive impact of this strategy on the third step of the HIV care cascade has recently been highlighted in four randomized population-based trials [36]. Furthermore, several studies have shown that ART initiation, including early initiation through the ‘test-and-treat’ strategy, leads to a significant decrease in sexual risk behaviors [24,37–39] thanks to greater opportunities for counselling and psychological support arising out of more frequent contact with HIV services [40]. Such a decrease may also contribute to lower the risk of HIV transmission.”

22. Line 324-326-emphazise and discuss supportive adherence counseling after ART initiation

As suggested, we provided more details about the implications of our findings regarding the low level of adherence to ART (see p.18, lines 314-318), as follows: 

“These findings reveal the huge challenges still facing HIV services in Cameroon to achieve the third ‘95’ UNAIDS target (i.e., VLS in 95% of PLHIV treated with ART by 2025) and to prevent new HIV infections. They also highlight the need to further develop supportive adherence services (e.g., peer-to-peer support, adherence clubs, and short message services) which have been shown to have a positive impact on adherence and viral suppression among PLHIV in LMIC [42].” 

23. Line 328-331-Discuss test and treat strategy to address HIV care and treatment cascade

As suggested, we outlined in the Discussion section the positive impact of the “test-and-treat” strategy on the two last steps of the HIV care cascade (access to ART and viral suppression) as follows (see p.18, lines 326-327): 

“In particular, the positive impact of this strategy on the third step of the HIV care cascade has recently been highlighted in four randomized population-based trials [36].”

However, as the implementation of this public health strategy results in a rapid and large increase in the number of ART-treated patients in HIV services, human resources shortages and inadequate ART supplies may be exacerbated when funding is not sufficient to face the increased demand. In our Limitations section (see p.20/21, lines 372-389), we therefore discussed the negative indirect effects of implementing “test-and-treat” on the delivery and quality of HIV care services. Two recent surveys in Cameroon indicated that the test-and-treat strategy may increase the risk for PLHIV of experiencing structural barriers to access HIV services (e.g., long waiting times, delay to initiate ART) and being loss-to-follow-up after early ART initiation [8,9]. These findings provide a more nuanced picture of the impact of the “test-and-treat” strategy on the HIV care continuum which is important to surface in order to better understand the critical role played by healthcare supply-related factors. 

24. Final discussion and recommendations can be more focused on the importance of test and treat, task shifting and ARV stocks management to prevent ARV stock outs

We revised the Discussion section by providing additional information on the test and treat strategy (see responses to comments #23). We also provided further details on how to best adapt ARV stocks management in HIV services using previous findings from Cameroon (see p.22, lines 403-405), as follows:

 “A previous study in Cameroon recommended implementing a set of harmonization and coordination procedures, as well as continuous quality control monitoring throughout the whole ARV supply chain to ensure permanent availability of these drugs for PLHIV [43].”

Reviewer #3.

25. In terms of content/form, the document seems to me to be well written. The subject addresses an issue never before addressed in Cameroon to the best of my knowledge and worthy of interest for improving the national response against HIV in Cameroon. The ethical aspects are not well described, but all the required authorizations have been obtained. The methodology is simple and clear and the conclusions are linked to it.

Thank you for your comment. Our Methods section has been revised with further details regarding the study design and data collection as well as with some ethical issues (see p.6/7, lines 112-129): 

“Briefly, eligible PLWH (≥21 years old and HIV diagnosed >3 months) attending an outpatient consultation in one of the 19 participating HIV services were randomly selected and informed about the study. Patients willing to participate provided written informed consent before data collection. First, a standardized medical questionnaire was completed during the consultation by healthcare providers. The following clinical data were obtained from patient examinations and retrospective medical files: dates of HIV diagnosis and ART initiation, WHO clinical stage of HIV infection at ART initiation and at the time of the study, CD4 count at ART initiation, drug regimen at the time of the study, body mass index and any history of tuberculosis and hepatitis B co-infection and related diagnosis date. Second, patients answered a face-to-face questionnaire administered in a private room by trained independent interviewers which collected data on demographic, socioeconomic, behavioral, psychosocial and domestic information. More specifically, a series of questions were asked on adherence to ART, perceived health and HIV-related stigma as well as alcohol consumption and sexual behaviors during the 12 months prior to the study (number of sexual partners, experience of transactional sex, frequency of sexual relationships, HIV status and condom use with the two most recent partners). The questionnaire is available as supplementary material in Fiorentino et al., 2021 [20]. Third, a blood sample was taken to measure HIV viral load (only for patients ART treated >6 months) and CD4 cell count. All blood samples were analyzed by a reference HIV laboratory in Yaoundé.” 

We also revised our ethical statement to provide additional details about the access to the data (see p.24), as follows: 

“This study was conducted in compliance with international and national regulations on ethics and research on people. It received administrative authorisation from the Ministry of Public Health in Cameroon and was approved by the Cameroonian National Ethics Committee (approval reference: 2013/08/349/L/CNERSH/SP). All participants were informed about the study’s objectives and its modalities and all provided written consent to participate. All individual data collected in the research were anonymized using a patient identification number; only this number was reported in the data collection tools and the databases used for analyses.”

 

References

1. Coulaud PJ, Protopopescu C, Ndiaye K, Baudoin M, Maradan G, Laurent C, et al. Individual and healthcare supply-related barriers to treatment initiation in HIV-positive patients enrolled in the Cameroonian antiretroviral treatment access programme. Health Policy Plan. 2021 Mar 26;36(2):137–48. 

2. National AIDS Control Committee. Cameroon: 2018-2022 National Strategic Plan for HIV/AIDS and STIs [Internet]. 2017 [cited 2021 Oct 19]. Available from: https://www.ilo.org/dyn/natlex/natlex4.detail?p_lang=en&p_isn=110883

3. Ministry of Public Health. National guideline for HIV prevention and care in Cameroon [Internet]. 2014 [cited 2019 Jan 15]. Available from: https://www.childrenandaids.org/sites/default/files/2017-05/Cameroon_National-Integrated-HIV-Guidelines2014.pdf

4. World Health Organization. Summary country profile for HIV/AIDS treatment scale-up: Cameroon [Internet]. 2005. Available from: http://www.ncbi.nlm.nih.gov/pubmed/21205377

5. UNAIDS. HIV estimates with uncertainty bounds 1990-2020 | UNAIDS [Internet]. 2021 [cited 2021 Nov 29]. Available from: https://www.unaids.org/en/resources/documents/2021/HIV_estimates_with_uncertainty_bounds_1990-present

6. UNAIDS. Prevailing against pandemics by putting people at the centre [Internet]. 2020 [cited 2021 Nov 2]. Available from: https://www.unaids.org/en/resources/documents/2020/prevailing-against-pandemics

7. Bärnighausen T, Bloom DE, Humair S. Human Resources for Treating HIV/AIDS: Are the Preventive Effects of Antiretroviral Treatment a Game Changer? PLoS One [Internet]. 2016 [cited 2019 Nov 21];11(10):e0163960. Available from: http://www.ncbi.nlm.nih.gov/pubmed/27716813

8. Ajeh R, Ekane H, Thomas EO, Dzudie A, Jules AN. Perceived patients’ satisfaction, barriers and implications on engagement in antiretroviral treatment services in Cameroon within the HIV Test and Treat context. Am J Public Heal Res [Internet]. 2019 [cited 2021 Nov 4];7(2):73–80. Available from: http://www.sciepub.com/AJPHR/abstract/10494

9. Awoh RA, Ekane HG, Dzudie A, Thomas EO, Adedimeji A, Jules AN. Implications of the human immunodeficiency virus test and treat strategy on antiretroviral treatment uptake and retention outcomes in Cameroon. Int J Community Med Public Heal. 2019 Oct 24;6(11):4716. 

10. UNAIDS. The western and central Africa catch-up plan - Putting HIV treatment on the fast-track by 2018 [Internet]. 2017 [cited 2018 Sep 13]. Available from: http://www.unaids.org/en/resources/documents/2017/WCA-catch-up-plan

11. Tchouwa GF, Eymard-Duvernay S, Cournil A, Lamare N, Serrano L, Butel C, et al. Nationwide Estimates of Viral Load Suppression and Acquired HIV Drug Resistance in Cameroon. EClinicalMedicine. 2018 Jul 1;1:21–7. 

12. UNAIDS. AIDSinfo [Internet]. UNAIDS. 2021 [cited 2018 Sep 14]. Available from: http://aidsinfo.unaids.org/

13. Nansseu JRN, Bigna JJR. Antiretroviral therapy related adverse effects: Can sub-Saharan Africa cope with the new “test and treat” policy of the World Health Organization? Vol. 6, Infectious Diseases of Poverty. BioMed Central Ltd.; 2017. p. 1–5. 

14. Global Burden of Disease Health Financing Collaborator Network JL, Haakenstad A, Micah A, Moses M, Abbafati C, Acharya P, et al. Spending on health and HIV/AIDS: domestic health spending and development assistance in 188 countries, 1995-2015. Lancet (London, England) [Internet]. 2018 May 5 [cited 2018 Aug 18];391(10132):1799–829. Available from: http://www.ncbi.nlm.nih.gov/pubmed/29678342

15. Atun R, Chang AY, Ogbuoji O, Silva S, Resch S, Hontelez J, et al. Long-term financing needs for HIV control in sub-Saharan Africa in 2015-2050: a modelling study. BMJ Open [Internet]. 2016 Mar 6 [cited 2019 Nov 21];6(3):e009656. Available from: http://www.ncbi.nlm.nih.gov/pubmed/26948960

16. Boyer S, Clerc I, Bonono C-R, Marcellin F, Bilé P-C, Ventelou B. Non-adherence to antiretroviral treatment and unplanned treatment interruption among people living with HIV/AIDS in Cameroon: Individual and healthcare supply-related factors. Soc Sci Med [Internet]. 2011 Apr [cited 2019 Nov 8];72(8):1383–92. Available from: http://www.ncbi.nlm.nih.gov/pubmed/21470734

17. Boyer S, Eboko F, Camara M, Abé C, Nguini MEO, Koulla-Shiro S, et al. Scaling up access to antiretroviral treatment for HIV infection: the impact of decentralization of healthcare delivery in Cameroon. AIDS [Internet]. 2010 Jan [cited 2019 Jan 14];24(Suppl 1):S5–15. Available from: http://www.ncbi.nlm.nih.gov/pubmed/20023440

18. Tong C, Suzan-Monti M, Sagaon-Teyssier L, Mimi M, Laurent C, Maradan G, et al. Treatment interruption in HIV-positive patients followed up in Cameroon’s antiretroviral treatment programme: individual and health care supply-related factors (ANRS-12288 EVOLCam survey). Trop Med Int Heal [Internet]. 2018 Mar 1 [cited 2019 Nov 23];23(3):315–26. Available from: http://doi.wiley.com/10.1111/tmi.13030

19. Liégeois F, Eymard-Duvernay S, Boyer S, Maradan G, Kouanfack C, Domyeum J, et al. Heterogeneity of virological suppression in the national antiretroviral programme of Cameroon (ANRS 12288 EVOLCAM). HIV Med [Internet]. 2019 Jan [cited 2019 Nov 23];20(1):38–46. Available from: http://www.ncbi.nlm.nih.gov/pubmed/30362279

20. Fiorentino M, Sow A, Sagaon-Teyssier L, Mora M, Mengue MT, Vidal L, et al. Intimate partner violence by men living with HIV in Cameroon: Prevalence, associated factors and implications for HIV transmission risk (ANRS-12288 EVOLCAM). PLoS One. 2021 Feb 1;16(2 Febuary). 

21. Bachanas P, Kidder D, Medley A, Pals SL, Carpenter D, Howard A, et al. Delivering Prevention Interventions to People Living with HIV in Clinical Care Settings: Results of a Cluster Randomized Trial in Kenya, Namibia, and Tanzania. AIDS Behav. 2016 Sep 1;20(9):2110–8. 

22. Okoboi S, Castelnuovo B, Moore DM, Musaazi J, Kambugu A, Birungi J, et al. Risky sexual behavior among patients onlong-term antiretroviral therapy: A prospective cohort study in urban and rural Uganda. AIDS Res Ther. 2018 Oct 19;15(1). 

23. Shuper PA, Kiene SM, Mahlase G, MacDonald S, Christie S, Cornman DH, et al. HIV Transmission Risk Behavior Among HIV-Positive Patients Receiving Antiretroviral Therapy in KwaZulu-Natal, South Africa. AIDS Behav. 2014 Aug 1;18(8):1532–40. 

24. Ndziessi G, Cohen J, Kouanfack C, Marcellin F, Carierri MP, Laborde-Balen G, et al. Susceptibility to transmitting HIV in patients initiating antiretroviral therapy in rural district hospitals in Cameroon (Stratall ANRS 12110/ESTHER trial). PLoS One. 2013 Apr 30;8(4). 

25. Jean K, Gabillard D, Moh R, Danel C, Fassassi R, Desgrées-Du-Loû A, et al. Effect of early antiretroviral therapy on sexual behaviors and HIV-1 transmission risk among adults with diverse heterosexual partnership statuses in côte d’ivoire. J Infect Dis. 2014;209(3):431–40. 

26. Siedner MJ, Musinguzi N, Tsai AC, Muzoora C, Kembabazi A, Weiser SD, et al. Treatment as long-term prevention: Sustained reduction in HIV sexual transmission risk with use of antiretroviral therapy in rural Uganda. AIDS. 2014 Jan 14;28(2):267–71. 

27. Eisinger RW, Dieffenbach CW, Fauci AS. HIV viral load and transmissibility of HIV infection undetectable equals untransmittable. Vol. 321, JAMA - Journal of the American Medical Association. American Medical Association; 2019. p. 451–2. 

28. Carrieri P, Cailleton V, Le Moing V, Spire B, Dellamonica P, Bouvet E, et al. The Dynamic of Adherence to Highly Active Antiretroviral Therapy: Results From the French National APROCO Cohort. JAIDS J Acquir Immune Defic Syndr [Internet]. 2001 Nov 1 [cited 2019 Nov 23];28(3):232–9. Available from: http://www.ncbi.nlm.nih.gov/pubmed/11694829

29. UNAIDS. Cameroon developing subnational estimates of HIV prevalence and the number of people living with HIV. 2014;19. Available from: http://www.unaids.org/en/resources/documents/2014/2014_subnationalestimatessurvey_cameroon

30. Ndawinz JDA, Chaix B, Koulla-Shiro S, Delaporte E, Okouda B, Abanda A, et al. Factors associated with late antiretroviral therapy initiation in Cameroon: a representative multilevel analysis. J Antimicrob Chemother [Internet]. 2013 Jun 1 [cited 2018 Jul 8];68(6):1388–99. Available from: http://www.ncbi.nlm.nih.gov/pubmed/23391713

31. Institut National de la Statistique (INS) et ICF. Enquête Démographique et de Santé du Cameroun 2018. Yaoundé, Cameroun et Rockville, Maryland, USA; 2020 Feb. 

32. Cohen MS, Chen YQ, McCauley M, Gamble T, Hosseinipour MC, Kumarasamy N, et al. Antiretroviral Therapy for the Prevention of HIV-1 Transmission. N Engl J Med [Internet]. 2016 Sep 18 [cited 2018 Aug 13];375(9):830–9. Available from: http://www.nejm.org/doi/10.1056/NEJMoa1600693

33. TEMPRANO ANRS 12136 Study Group, Danel C, Moh R, Gabillard D, Badje A, Le Carrou J, et al. A Trial of Early Antiretrovirals and Isoniazid Preventive Therapy in Africa. N Engl J Med [Internet]. 2015 Aug 27 [cited 2018 Jul 9];373(9):808–22. Available from: http://www.ncbi.nlm.nih.gov/pubmed/26193126

34. INSIGHT START Study Group, Lundgren JD, Babiker AG, Gordin F, Emery S, Grund B, et al. Initiation of Antiretroviral Therapy in Early Asymptomatic HIV Infection. N Engl J Med [Internet]. 2015 Aug 27 [cited 2018 Sep 18];373(9):795–807. Available from: http://www.ncbi.nlm.nih.gov/pubmed/26192873

35. World Health Organization. Guidelines Guideline on When To Start Antiretroviral Therapy and on Pre-Exposure Prophylaxis for Hiv. World Heal Organ. 2015; 

36. Havlir D, Lockman S, Ayles H, Larmarange J, Chamie G, Gaolathe T, et al. What do the Universal Test and Treat trials tell us about the path to HIV epidemic control? J Int AIDS Soc [Internet]. 2020 Feb 24 [cited 2021 Nov 30];23(2):e25455. Available from: https://onlinelibrary.wiley.com/doi/10.1002/jia2.25455

37. Jean K, Gabillard D, Moh R, Danel C, Desgrées-du-Loû A, N’takpe JB, et al. Decrease in sexual risk behaviours after early initiation of antiretroviral therapy: A 24-month prospective study in Côte d’Ivoire. J Int AIDS Soc. 2014 Jun 30;17(1). 

38. Wandera B, Kamya MR, Castelnuovo B, Kiragga A, Kambugu A, Wanyama JN, et al. Sexual behaviors over a 3-year period among individuals with advanced HIV/AIDS receiving antiretroviral therapy in an Urban HIV clinic in Kampala, Uganda. J Acquir Immune Defic Syndr. 2011 May 1;57(1):62–8. 

39. Ndziessi G, Cohen J, Kouanfack C, Boyer S, Moatti JP, Marcellin F, et al. Changes in sexual activity and risk behaviors among PLWHA initiating ART in rural district hospitals in Cameroon - Data from the STRATALL ANRS 12110/ESTHER trial. AIDS Care - Psychol Socio-Medical Asp AIDS/HIV. 2013 Mar 1;25(3):347–55. 

40. Protopopescu C, Marcellin F, Préau M, Gabillard D, Moh R, Minga A, et al. Psychosocial correlates of inconsistent condom use among HIV-infected patients enrolled in a structured ART interruptions trial in Côte d’Ivoire: Results from the TRIVACAN trial (ANRS 1269): Short Communication. Trop Med Int Heal. 2010 Jun;15(6):706–12. 

41. PEPFAR. Cameroon Country Operational Plan (COP) 2017 Strategic Direction Summary [Internet]. 2017. Available from: https://www.state.gov/wp-content/uploads/2019/08/Cameroon-14.pdf

42. Haberer JE, Sabin L, Amico KR, Orrell C, Galárraga O, Tsai AC, et al. Improving antiretroviral therapy adherence in resource-limited settings at scale: a discussion of interventions and recommendations. J Int AIDS Soc [Internet]. 2017 Jan 1 [cited 2021 Nov 4];20(1):21371. Available from: http://doi.wiley.com/10.7448/IAS.20.1.21371

43. Djobet MPN, Singhe D, Lohoue J, Kuaban C, Ngogang J, Tambo E. Antiretroviral therapy supply chain quality control and assurance in improving people living with HIV therapeutic outcomes in Cameroon. AIDS Res Ther. 2017 Apr 4;14(1):1–8.

---

## [Decision Letter · Decision Letter 1]

22 Mar 2022

Individual and healthcare supply-related HIV transmission factors in HIV-positive patients enrolled in the antiretroviral treatment access program in the Centre and Littoral regions in Cameroon (ANRS-12288 EVOLCam survey)

PONE-D-21-22603R1

Dear Dr. Coulaud,

We’re pleased to inform you that your manuscript has been judged scientifically suitable for publication and will be formally accepted for publication once it meets all outstanding technical requirements.

Kind regards,

Miquel Vall-llosera Camps

Senior Editor

PLOS ONE

Reviewers' comments:

Reviewer's Responses to Questions

**Comments to the Author**

1. If the authors have adequately addressed your comments raised in a previous round of review and you feel that this manuscript is now acceptable for publication, you may indicate that here to bypass the “Comments to the Author” section, enter your conflict of interest statement in the “Confidential to Editor” section, and submit your "Accept" recommendation.

Reviewer #1: All comments have been addressed

Reviewer #2: All comments have been addressed

Reviewer #3: All comments have been addressed

2. Is the manuscript technically sound, and do the data support the conclusions?

Reviewer #1: Yes

Reviewer #2: Yes

Reviewer #3: Yes

3. Has the statistical analysis been performed appropriately and rigorously? 

Reviewer #1: Yes

Reviewer #2: Yes

Reviewer #3: Yes

4. Have the authors made all data underlying the findings in their manuscript fully available?

Reviewer #1: No

Reviewer #2: Yes

Reviewer #3: Yes

5. Is the manuscript presented in an intelligible fashion and written in standard English?

Reviewer #1: Yes

Reviewer #2: Yes

Reviewer #3: Yes

6. Review Comments to the Author

Reviewer #1: Affiliation N0 4 ; instead of «Faculté de Médecine et de sciences biomédicales », please modify it as «Faculté de Médecine et de Sciences Biomédicales»

Reviewer #2: The findings of this study shall surely be usefuf at implementation level to balance the benetifs and risk of test and treat strategy on HIV risk transmission including effects on the health system supply and overall quality of services

Reviewer #3: all my comments have been taken into account

in particular the ethical considerations have been reviewed with the necessary details.

All individual data collected in the research were

anonymized using a patient identification number; only this number was reported in the data

collection tools and the databases used for analyses.

7. PLOS authors have the option to publish the peer review history of their article (what does this mean?). If published, this will include your full peer review and any attached files.

Reviewer #1: **Yes: **Georges Nguefack-Tsague

Reviewer #2: **Yes: **Anne Esther Njom Nlend

Reviewer #3: No

---

## [Editor Report · Acceptance letter]

28 Mar 2022

PONE-D-21-22603R1 

Individual and healthcare supply-related HIV transmission factors in HIV-positive patients enrolled in the antiretroviral treatment access program in the Centre and Littoral regions in Cameroon (ANRS-12288 EVOLCam survey) 

Dear Dr. Coulaud:

I'm pleased to inform you that your manuscript has been deemed suitable for publication in PLOS ONE. Congratulations! Your manuscript is now with our production department. 

Kind regards, 

on behalf of

Dr. Miquel Vall-llosera Camps 

Staff Editor

PLOS ONE